# Xenon plasma focused ion beam lamella fabrication on high-pressure frozen specimens for structural cell biology

Casper Berger [1,5], Helena Watson [1,2,5], James H. Naismith[1,3], Maud Dumoux [1] & Michael Grange [1,4] ✉

Cryo focused ion beam lamella preparation is a potent tool for in situ structural biology, enabling the study of macromolecules in their native cellular environments. However, throughput is currently limited, especially for thicker, more biologically complex samples. We describe how xenon plasma focused ion beam milling can be used for routine bulk milling of thicker, high-pressure frozen samples. We demonstrate lamellae preparation with a high success rate on these samples and determine a 4.0 Å structure of the *Escherichia coli* ribosome on these lamellae using sub volume averaging. We determine the effects on sample integrity of increased ion currents up to 60 nA during bulk milling of thicker planar samples, showing no measurable damage to macromolecules beyond an amorphous layer on the backside of the lamellae. The use of xenon results in substantial structural damage to particles up to approximately 30 nm in depth from the milled surfaces, and the effects of damage become negligibly small by 45 nm. Our results outline how the use of high currents using xenon plasma focused ion beam milling may be integrated into FIB milling regimes for preparing thin lamellae for high-resolution in situ structural biology.

Cryo-electron tomography (cryo-ET), combined with focused ion beam (FIB) lamella fabrication, has become an effective method to determine how macromolecular structure and biological function are linked in the cell[1,2]. Advances in automated lamella fabrication[3–5], cryo-electron tomography data collection[6–8] and sub-tomogram averaging (STA)[9,10] have enabled structures of large macromolecules present at high cellular concentrations to be determined at pseudo-atomic resolution[2,11–14]. Further improvements in quality and throughput of complex lamella sample preparation approaches would benefit in situ structural biology by resulting in an increase in the attainable resolution for abundant macromolecules and bringing rarer macromolecules within scope.

The limitations of gallium liquid metal ion sources for FIB lamella fabrication have been previously discussed in the context of material science applications[15–17]. In summary, the limited current density of these sources significantly limits the rate of bulk milling. This is a significant drawback for production from thicker, more biologically complex samples such as tissues and small organisms[18]. Plasma sources however remain collimated even at high currents[17] enabling the use of higher current regimes (up to 2.5 μA); the effect upon biological samples remains to be determined. Plasma ion sources have been demonstrated as an effective alternative to gallium sources for lamella production from mammalian cells[2] and protein crystals[19,20]. Xenon has a higher milling rate on vitreous samples compared to other ion sources[2,21], which in combination with the range of currents possible for plasma focused ion beams (PFIB) may enable high-throughput milling strategies for thicker biological samples such as serial lift-out[18,22] and the 'Waffle' method[23].

[1]Structural Biology, The Rosalind Franklin Institute, Harwell Science & Innovation Campus, Didcot OX11 0QS, UK. [2]School of Molecular Biosciences, College of Medical Veterinary and Life Sciences, University of Glasgow, Glasgow, UK. [3]Mathematical, Physical and Life Sciences Division, University of Oxford, Oxford, UK. [4]Division of Structural Biology, Wellcome Centre for Human Genetics, University of Oxford, OX3 7BN Oxford, United Kingdom. [5]These authors contributed equally: Casper Berger, Helena Watson. ✉e-mail: michael.grange@rfi.ac.uk

FIB milling is known to damage materials due to ion impacts on the milled surfaces and the resulting collision cascade[15,16,24,25]. For materials science applications, the main determinants of the depth and extent of the damage penetration are known to be the composition of the material being milled, the ion source, ion incident angle and ion accelerating voltage[15,16,24]. Using in situ sub-tomogram averaging and B-factor analysis[26], it was demonstrated that 30 kV argon PFIB milling on biological samples results in damage characterised by reduced information content at depths up to 30 to 45 nm from the milling surfaces[2]. The damage penetration for 8 kV and 30 kV gallium FIB milling has been characterised[27–29], reporting penetration depths for the damage between 30 and 60 nm, with reduced depth of damage at lower voltage FIB milling.

Here, we present a workflow for the use of xenon PFIB milling to prepare lamellae from ~25 μm thick high-pressure frozen biological samples with currents up to 60 nA at 30 kV. The use of high-pressure frozen samples has broad potential as this extends the applications that may be harnessed via cryo-ET[23]. With this method we determine the structure of the *Escherichia coli* ribosome to a resolution of 4.0 Å. We show that xenon plasma milling using this protocol results in damage to lamellae which has a substantial effect on structural information up to 30 nm from the milling surfaces and negligible impact by depths of approximately 45 nm, and also describe the effects of the use of a 60 nA probe for bulk milling. Our results demonstrate that xenon plasma milling with currents up to 60 nA can be integrated into milling workflows for high-resolution in situ structural biology.

## Results

### Xenon plasma FIB lamella preparation of high-pressure frozen samples

We adapted previously published methods for lamella fabrication of high-pressure frozen samples[23] to plasma FIB milling. Our method enabled simultaneous site preparation, with benefits in throughput and the removal of ice contamination (Fig. 1, Table 1). We high-pressure froze *Escherichia coli* and *Saccharomyces cerevisiae* in solution, using an electron microscopy grid without support film as a spacer between two flat-sided planchettes to obtain vitreous samples with a thickness of ~25 μm (the thickness of the grid). Ice contamination ranging in size from a few to hundreds of μm was frequently seen on the samples. We therefore added an ice contamination removal step where both sides of the grid are imaged with the PFIB beam at low magnification with currents of 4–60 nA for short periods of time (Fig. 1a–c, Table 1, Supplementary Video 1). To effectively produce lamellae from material that is ~25 μm thick on a square mesh grid, it is essential to remove material at each side of the intended position effectively. After GIS deposition, two trenches per lamella site are milled 90° relative to the grid from the backside of the grid based on the stage limits; the front of the grid is not accessible at a perpendicular angle with the PFIB of the microscope used here (Fig. 1d, e). Xenon PFIB milling has been demonstrated to give higher milling rates for vitrified samples compared to other ion sources[2,21]. We used xenon at 60 nA for trench milling (compared to 15 nA described by Kelley et al.[23]). Next, sites are selected for their surface smoothness and material is removed below the target positions from the front of the grid at increasingly shallower angles towards the grid surface (45°, 28° and 20° relative to the grid) (Fig. 1d). This leads to progressive removal of the material underneath while not being constrained by the grid bar or the material at the front. To further accelerate trench preparation, we used a routine that utilised a low magnification in the ion beam view to simultaneously prepare multiple trenches within the field of view (Fig. 1e, f). Afterwards, rough, medium and fine automated milling steps were performed using progressively lower currents (Fig. 1g–k and n–q), followed by optional manual notch pattern milling for

stress relief (Fig. 1l, r) and automated final lamellae polishing (Fig. 1m, s).

### Xenon plasma milling of high-pressure frozen samples results in high-quality lamella for STA

We applied our adapted milling method to high-pressure frozen *E. coli* and *S. cerevisiae* and found that lamellae could be reliably prepared with success rates of 70-84% (Fig. 2a, Supplementary Fig. 1, Supplementary Fig. 2). We measured the local lamella thickness in tomograms suitable for STA and found average thickness values of 144–209 nm (Fig. 2b–f; Supplementary Video 2) with lower thickness values at the front compared to the back of the lamellae (Supplementary Fig. 3).

We extracted ribosomes from tomograms in the *E. coli* dataset after benchmarking different filters for automated particle picking (Supplementary Fig. 4; see Materials and Methods), resulting in identification of 224,823 predicted ribosome particles. Classification and refinement allowed determination of a consensus 70S ribosome structure with a global resolution of 4.0 Å (Supplementary Fig. 5, Supplementary Fig. 6), with large regions of the structure reaching a resolution at the Nyquist sampling limit of 3.8 Å (Fig. 2g, Supplementary Video 3). Indicative of the quality of the map, amino acid side chains could be readily identified in the structure from both small and large ribosomal subunits (Fig. 2h). This 4.0 Å *E. coli* ribosome structure demonstrates the use of higher current plasma regimes (i.e. incorporating 60 nA steps) for lamella preparation for high-resolution in situ structural biology.

### Damage to the backside of lamellae

We observed a region without any apparent biological features on the backside of each lamella (Fig. 3a). This area consists of a striated layer (Fig. 3b, Supplementary Videos 4 and 5), followed by an amorphous region with a mean length of 0.72 μm (SD; 0.22 μm, $n = 15$) (Fig. 3b, c, Supplementary Fig. 7a). The areas directly adjacent to the amorphous area do contain biological features including membrane bilayers and ribosomes (Fig. 3c, d). We did not observe Bragg reflections in tilt-series recorded on the amorphous layer, indicating that it remains vitreous (Supplementary Video 4). We speculated that the striated and amorphous regions may have been caused by the high PFIB current used during the perpendicular trench milling step (Supplementary Fig. 8). We therefore prepared lamellae using 4 nA or 60 nA for the first trench milling step and compared the length of the amorphous region (Supplementary Fig. 7b–d). We found that the amorphous area was present in both, but significantly shorter in the 4 nA lamellae (4 nA: mean SD; $0.5 \pm 0.07$ μm, $n = 5$; 60 nA: mean SD; $0.79 \pm 0.18$ μm, $n = 21$; $p = 0.0012$).

To determine whether damage is present in regions of the lamellae adjacent to the amorphous layer, we carried out structural studies using B-factor analysis on the sample milled with 60 nA. In electron microscopy, B-factor plots show the relationship between the number of particles used for reconstruction and the resulting resolution, and are sensitive to all factors that influence this, such as optics and sample characteristics[26]. We determined the B-factors of ribosomes located at different distances from the boundary between the amorphous region and the rest of the lamella. The distance of each ribosome from this boundary was calculated by using the distance of each tilt series acquisition area from the boundary, then correcting per-particle based on the refined coordinates of the ribosomes within each tomogram (Fig. 3e). Particles were then grouped by distance in groups of 1 μm from the boundary, and the B-factor calculated for each group up to a total distance of 5 μm (Fig. 3f, Table 2, Supplementary Fig. 9). The B-factors only ranged between 284 and 321 Å² across the groups, and no significant correlation was observed (Spearman correlation: $r = -0.4$, p-value = 0.50) between B-factors and distance from the boundary between the amorphous layer and the rest of the lamella (Supplementary Fig. 9). At the micron scale, we conclude there was no

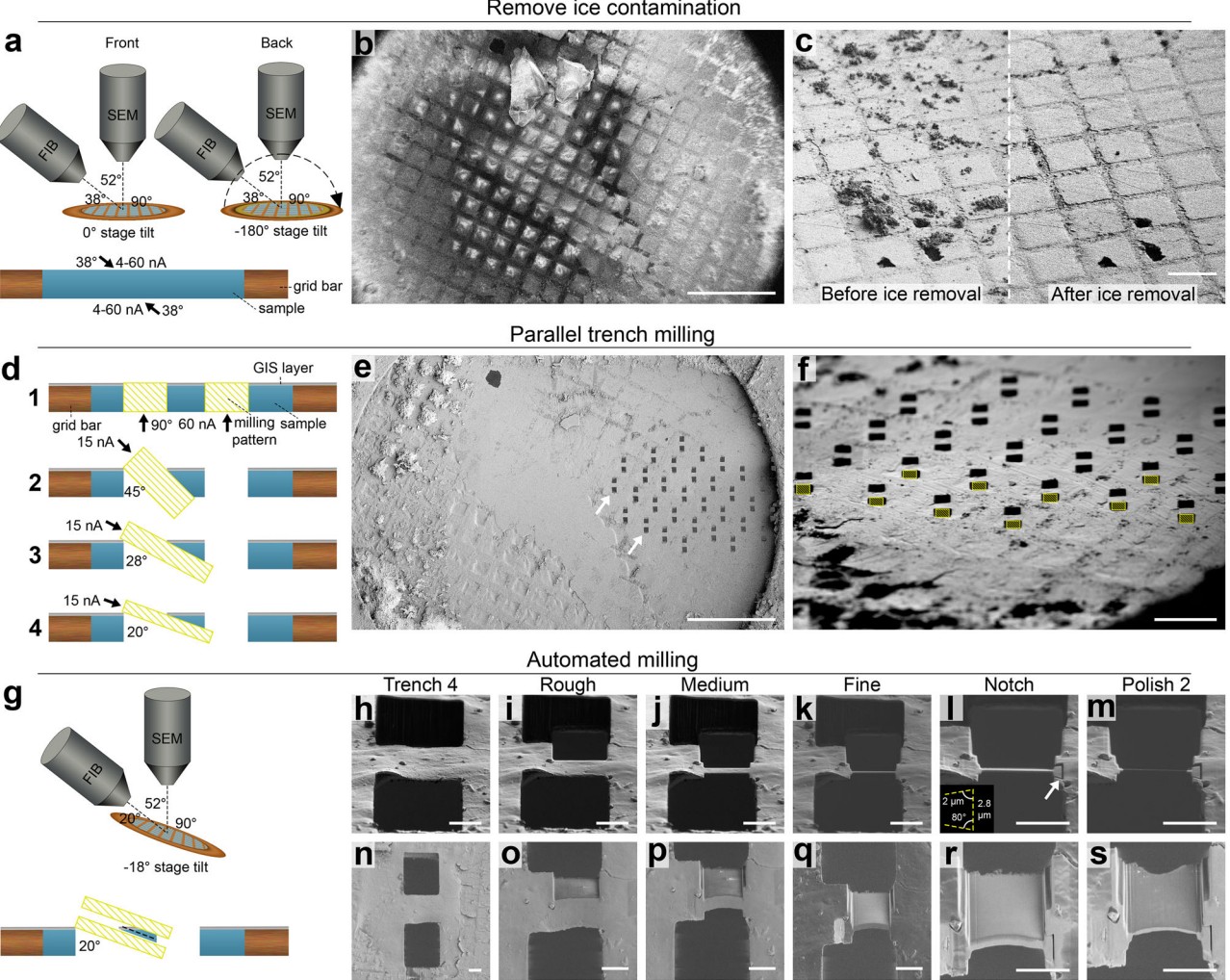

**Fig. 1 | Overview of xenon plasma FIB milling workflow for high-pressure frozen samples. a** Orientation of the grid relative to the PFIB and scanning electron microscope (SEM) during ice contamination removal using xenon beam on an Arctis PFIB/SEM from the front (left) and back (right) of the grid. Bottom: schematic orientation of a grid square and the angle and currents used to remove the ice contamination. **b** SEM grid overview before ice contamination removal. Scalebar: 500 μm. *n* = 3 whole grids. **c** PFIB images before (left) and after (right) ice contamination removal by PFIB imaging. Scalebar: 100 μm. **d** Schematic overview of milling pattern placement (yellow striped rectangles) during the four trench milling steps. **e** SEM overview of the grid shown in **b**, after ice contamination removal, GIS deposition and trench milling steps (example milled sites marked with white arrows). Using a low magnification ion beam, all trenches are milled at the same time. Typically, more trenches are prepared than required, and sites with high front surface roughness are excluded from automated milling. Scalebar: 500 μm. **f** PFIB

image of pattern placement for the 4th trench milling step. Milling patterns (yellow striped rectangles) are concurrently placed and milled in in horizontal rows, as tilting the grid varies the distance to the PFIB, resulting in a vertical focus gradient. Scalebar: 200 μm. **g** (top) Schematic overview of the grid orientation during automated milling relative to the PFIB and SEM beam. (bottom) Schematic overview of milling pattern placement relative to the forming lamella (black dashed line) during automated lamella preparation. **h–s** PFIB (top row) and SEM (bottom row) images of a lamella after the milling step indicated at the top. After the 4th trench milling step (**h, n**), automated lamella preparation for rough, medium and fine milling (**i–k** and **o–q**) at 4 nA, 1 nA and 0.3 nA respectively. After manually preparing notches at 0.1 nA (white arrow and bottom left of **l**) for stress-relief (**l, r**), two automated milling steps are performed at 30 pA (not shown) and 10 pA (**m, s**). Scalebars: 10 μm. *n* = 52 lamellae.

detectable damage propagation beyond the boundary of the amorphous region.

## Xenon PFIB surface damage penetrates lamellae to depths of 30 to 45 nm

We quantified the depth of ion beam damage propagation from the milling surface into the lamellae using B-factor analysis. B-factors were determined for ribosomes grouped by distance from the PFIB milling surfaces (Fig. 4a, Table 3). Higher B-factors (Fig. 4b) and lower resolutions for the same number of particles (Fig. 4c) were observed for ribosomes closer to the milling surfaces compared to those located deeper into the lamellae. To correct for factors that affect the attainable resolutions and B-factors of particle subsets (e.g. local lamella thickness and motion), matched controls for each distance group were created by randomly taking the same numbers of ribosomes from the same tomograms but at greater distances from the milling surfaces (Fig. 4a, Supplementary Fig. 10). We plotted the ratio of B-factors from a given depth group to its matched control and fitted an exponential decay curve (Fig. 4d). We also plotted the resolution losses (i.e. the difference in resolution between a group and its matched control) extrapolated from the average B-factors at each depth group (Fig. 4e). In the first 10 nm from the surface boundary models, large differences in B-factors and resolution between the distance groups and their matched controls are observed, which is likely a combined effect from partially ablating ribosomes (diameter of ~21 nm) as well as damage from ion impacts and the resulting collision cascade. This damage substantially reduces the information content of particles within 30 nm from the

**Table 1 | Milling parameters for xenon PFIB milling of high-pressure frozen samples**

| Step | FIB angle relative to grid | Pattern width | Pattern height/ overlap (%) | Offset from lamella[a] | Pattern type | Milling time[b] | Drift correction interval | Ion beam current |
|---|---|---|---|---|---|---|---|---|
| Ice cleaning front | 38° | 980 µm by 653 µm | - | - | Imaging | ~30–90 s | - | 4, 15 and/ or 60 nA |
| Ice cleaning back | 38° | 980 µm by 653 µm | - | - | Imaging | ~30–90 s | - | 4, 15 and/ or 60 nA |
| Trench milling 1 | 90° | 25 µm | 30 µm | 25 µm | Cleaning cross section | 3 m:00 s | - | 60 nA |
| Trench milling 2 to 4 | 45°, 28° and 20° | 22 µm | 18 to 22 µm | - | Rectangular milling pattern | 1 m:48 s to 2 m:12 s | - | 15 nA |
| Rough milling | 20° | 20 µm[c] (top) 19 µm (bottom) | 8 µm | 6 µm | Rectangular milling pattern | 7 m:45 s | 500 s | 4 nA |
| Medium milling | 20° | 17.3 µm[c] (top) 17 µm (bottom) | 3 µm[a] 250% | 1.6 µm | Rectangular milling pattern | 7 m:45 s | 500 s | 1 nA |
| Fine milling | 20° | 16.7 µm[c] (top) 16.2 µm (bottom) | 0.8 µm[a] 240% | 700 nm | Rectangular milling pattern | 7 m:24 s | 60 s | 0.3 nA |
| Notch | 20° | - | - | - | Line pattern | 50 s | - | 0.1 nA |
| Polishing 1 | 20° | 16 µm[c] | 500 nm[a] 200% | 120 nm | Rectangular milling pattern | 10 m:15 s | 60 s | 30 pA |
| Polishing 2 | 20° | 16 µm[c] | 120 nm[a] 200% | 0 nm | Rectangular milling pattern | 7 m:52 s | 30 s | 10 pA |

[a]Approximate pattern heights were measured from the placed patterns, as AutoTEM does not display the absolute values. A target thickness as defined in AutoTEM software of 120 nm was used. AutoTEM adds an additional offset in the distance between how the patterns are place from the target thickness. For a target thickness of 120 nm at 10 pA with xenon, the distance between the milling patterns is ~245 nm.
[b]For all patterns from one lamella site, but for ice cleaning per grid.
[c]Typical width of final lamella is 16 µm but can be adapted depending on the lamella site.

milling surfaces, and the damage effects become negligibly small within 30 to 45 nm from the surfaces.

To determine the overall impact of the surface damage on STA resolution, we performed B-factor analyses for three sets of 10,000 randomly selected particles. One set has particles more than 30 nm from the milling surfaces (minimal PFIB surface damage), the second set has particles less than 30 nm from the milling surfaces (substantial PFIB surface damage) and the third set has no distance constraint. STA of particles within 30 nm of the PFIB milling surfaces results in lower resolution structures compared to the other two sets (Fig. 4f).

## Discussion

We have adapted previously published methods for milling high-pressure frozen samples[23] with a xenon PFIB source, improving throughput and implementing an ice contamination reduction step. We can routinely prepare ~15–20 lamellae suitable for cryo-ET from high-pressure frozen samples in a 24-hour cryo-PFIB/SEM session, providing comparable success rate, throughput, and lamellae thickness to current methods for plasma- and gallium-milled plunge-frozen cells[2,3]. This is due to the combination of xenon's higher milling rate compared to other ion beam sources, the use of higher currents than previously reported for lamella preparation, and setting up multiple trench sites per FIB field of view. Even higher currents would further accelerate trench preparation, but samples of this thickness remain rate-limited by the final thinning of lamellae to electron transparency. Higher currents may instead primarily benefit thicker samples such as tissues, which require ablation of much greater volumes of material. For these significantly thicker samples, alternative milling strategies such as serial lift-out will be necessary, however the throughput of a 'waffle-style' approach remains substantially higher for samples where lift-out is not required[22]. To demonstrate the quality of the lamellae produced by our approach we determined a 4.0 Å resolution structure of the *E. coli* 70S ribosome.

One feature of our approach is the use of the PFIB to reduce the surface ice contamination from the high-pressure freezing process; this could have the potential to damage the sample. However, we believe this to be unlikely as the large field of view (980 µm × 653 µm) means the estimated milling rate is ~850 times lower than in the 60 nA trench milling step (25 µm × 30 µm area), and surface damage from the ion collision cascade should be restricted to the front 100–300 nm of the lamella length. The reduction of surface ice allows high quality data collection from grids which would otherwise have been abandoned, with consequent reduction in throughput. The ability to "rescue" grids will particularly benefit projects with complex biological sample preparation where loss of a specimen represents significant experimental effort. We hypothesise that the mechanism for ice contamination removal could be due to changes in charging conditions of the ice due to exposure to ions, and can likely also be used with other ion species.

Interestingly, we observed striated and amorphous regions only on the back side of lamellae. We suggest that this region on the backside likely resulted from the perpendicular trench milling step, as the positioning of the lamella site (i.e. typically offset from the front trench by several microns) and the shallow milling angles used (Supplementary Fig. 8) means that only the lamella back side intersects a trench side wall. No Bragg reflections were seen in tilt-series collected either in the amorphous region or the rest of the lamellae, indicating that the use of higher currents up to 60 nA has no negative effect on sample vitrification despite previous concerns[30]. B-factor analysis revealed no detectable effects of proximity to the back of the lamellae on the micron scale, indicating suitability for high resolution STA. We therefore currently see no reason to avoid acquiring data adjacent to this region.

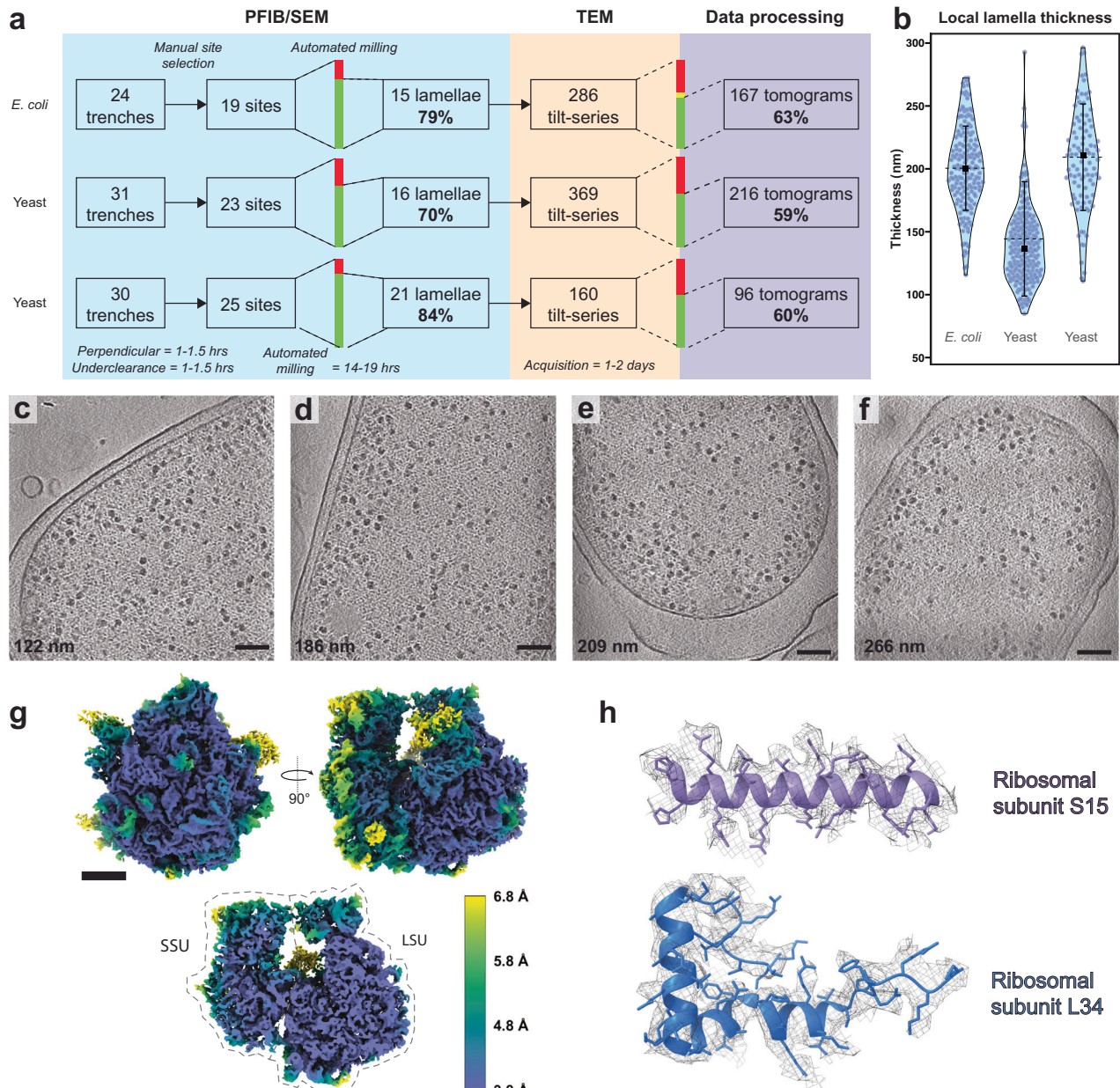

**Fig. 2 | Overview of lamella quality and _E. coli_ ribosome STA. a** Schematic overview of lamella fabrication workflow showing the throughput of trench preparation, selection for automated lamellae preparation, cryo-electron tomography tilt-series acquisition and selection of reconstructed tomograms for further analysis (see materials and methods for selection criteria). **b** Violin plots for the local lamella thickness distribution of the tomograms selected as suitable for analysis in (**a**) for the _E. coli_ (left) and two _S. cerevisiae_ (middle and right) datasets (L-R: _n_ = 167, 216, 96 technical replicates, respectively). The median values are indicated with a black square (L-R: 200, 137, 211 nm), the mean indicated with a black horizontal line (L-R: 201, 144, 209 nm) and standard deviations with vertical error bars (L-R: ±34,

±45, ±42 nm). Source data are provided as a Source Data File. **c–f** Examples of reconstructed slices of _E. coli_ tomograms, recorded on lamellae with local thickness values of 122, 186, 209 and 266 nm respectively. Scale bars: 100 nm. Slices are representative of _n_ = 167 _E. coli_ tomograms. **g** Consensus STA density map of surfaces (top) and central slice (bottom) of 70S ribosome, coloured by local resolution. The small ribosomal subunit (SSU) and large ribosomal subunit (LSU) are indicated. Scale bar: 5 nm. **h** Representative model (PDB: 6ORE[45]) fits into the electron density map for the following ribosomal subunit chains: (L-R) SSU subunit S15, LSU subunit L34.

We cannot unambiguously identify the cause for the formation of the amorphous layer. Lamellae prepared from trenches milled at 4 nA or 60 nA both have an amorphous region on the back, although it is significantly shorter for trenches prepared at 4 nA (Supplementary Fig. 7). Both redeposition and amorphisation effects depend on the sputter rate, which is influenced by both beam current and beam angle incident to the surface[31,32]. As trench milling is performed with a 60 nA current and perpendicular to the grid surface, both effects could be possible causes for the formation of this amorphous layer. Since the back of the grid is not coated with a protective platinum layer, redeposited material from milling through the back would primarily be derived from the biological sample, which would explain the similar contrast observed in the amorphous layer. We also speculate that the type of milling pattern (e.g. cleaning cross-section or rectangular milling pattern) may affect the morphology of this region as this also affects the timeframe in which local regions of sample are exposed to ion beam energy. The mechanism of formation of this region remains to be

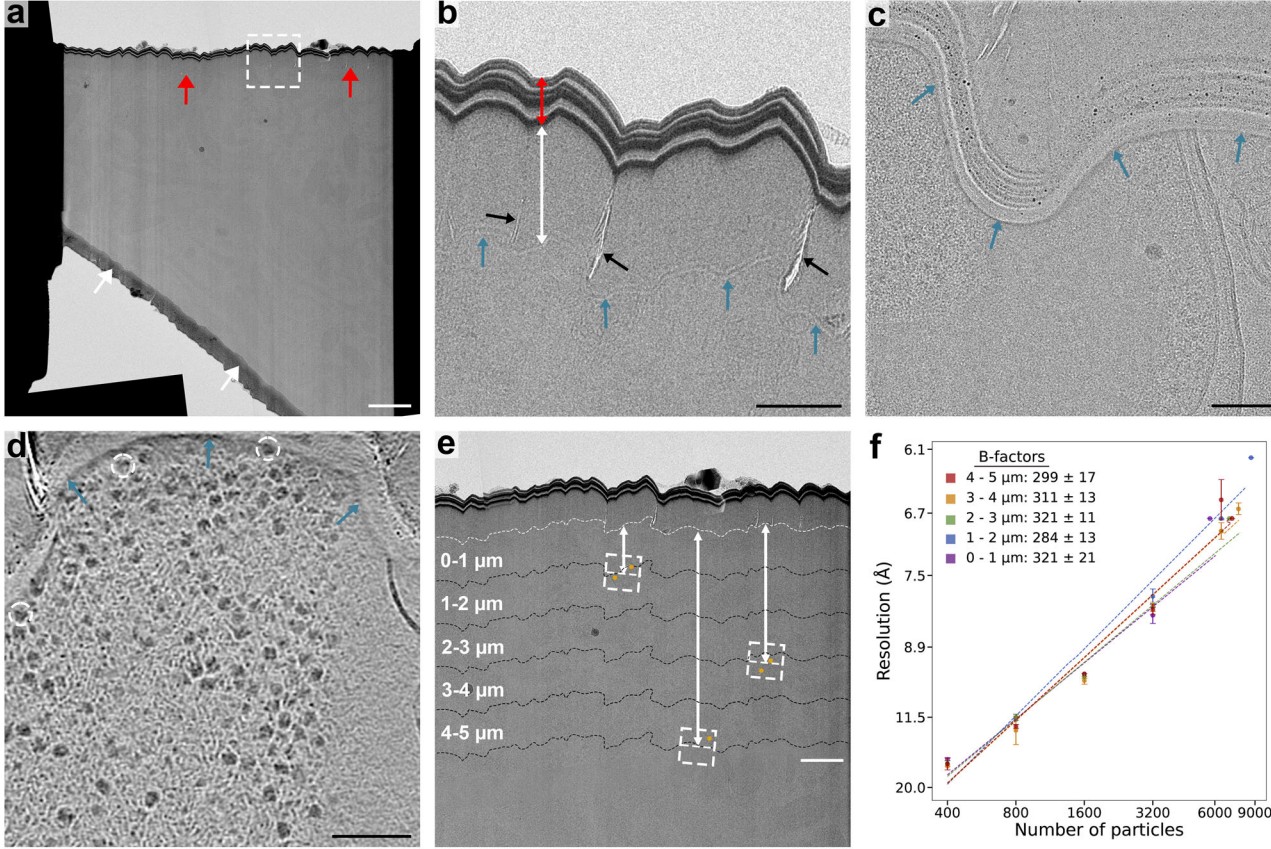

**Fig. 3 | Effects of backside amorphous and striated areas on STA resolution.**
**a** TEM overview of a lamella prepared from high-pressure frozen *E. coli* described in Fig. 1 (*n* = 52 lamellae). Amorphous areas are visible only on the backside of the lamella (red arrows), not on the front near the GIS layer (white arrows). White dashed square indicates area shown in (**b**). Scalebar: 2 µm. **b** TEM image of the amorphous area on the backside of the lamella, in the area shown in the white dashed square in (**a**). A distinct striated pattern of alternating high- and low electron-dense material is visible at the back of the lamella (red double-headed arrow), with an area without any remaining biological contrast of typically 0.5 to 1.5 µm in length (white double-headed arrow), which often contains small cracks (black arrows). The area without biological contrast is delineated by a striated pattern (blue arrows), weaker in contrast than at the very back of the lamella. Scalebar: 500 nm. **c** Tilt-image of a bacterium, where the contrast of the bacterium abruptly disappears at the border of amorphous area without biological contrast

(blue arrows). Scalebar: 100 nm. *n* = 15. **d** Tomographic slice of a bacterium near the area without biological contrast (blue arrows), with ribosomes clearly visible directly adjacent to this border (white dashed circles). Scalebar: 100 nm. *n* = 15. **e** Enlarged TEM overview of the lamella shown in (**a**). For each ribosome (orange circle) the distance to the amorphous layer (white dashed line) was determined by taking the distance of the corresponding tomogram (white dashed squares) to the amorphous layer and offsetting it with the ribosome position in the tomogram. Ribosomes were then grouped based on their distance to the amorphous layer in 1 µm bins (black dashed lines). Scalebar: 1 µm. **f** B-factor plot for all ribosome distance groups from the amorphous layer on the backside of the lamellae, in 1 µm groups. Resolution is plotted on square inverse scale and particle number is plotted on log$_e$ scale, with resolutions reported as mean ± SE (*n* = 3 technical replicates of the B-factor analysis). Source data are provided as a Source Data File.

## Table 2 | B-factor analysis of backside damage

| Backside distance group | Number of particles per distance group | Number of tomograms per distance group | Average lamella thickness (nm)[a] | B-factor (Å²)[b] | Global resolution (Å)[c] |
|---|---|---|---|---|---|
| <1 µm | 5702 | 18 | 213 ± 29 | 321 ± 21 | 6.8 |
| 1–2 µm | 8655 | 26 | 214 ± 28 | 284 ± 13 | 6.2 |
| 2–3 µm | 6941 | 21 | 216 ± 27 | 321 ± 11 | 6.8 |
| 3–4 µm | 7623 | 26 | 205 ± 30 | 311 ± 13 | 6.6 |
| 4–5 µm | 7146 | 25 | 195 ± 25 | 299 ± 17 | 6.8 |

[a]Average lamella thickness for each particle per distance group.
[b]Reported as mean ± SE.
[c]For one of *n* = 3 repeats.

determined and will require future experimentation to deconvolve the contributing factors.

Given the stochastic nature of surface damage propagation from ion-sample interactions, the effects of ion beam damage on structural information are expected to decay exponentially with increasing distance from lamella surfaces, thus caution is required in defining a discrete limit to the depth of damage penetration. We therefore use B-factor ratios between each depth group and its corresponding matched control to describe the damage curve. With xenon plasma ion beam milling, there was significant reduction in structural quality up to 30 nm from the surface, the effect of which becomes negligibly small by depths of up to approximately 45 nm

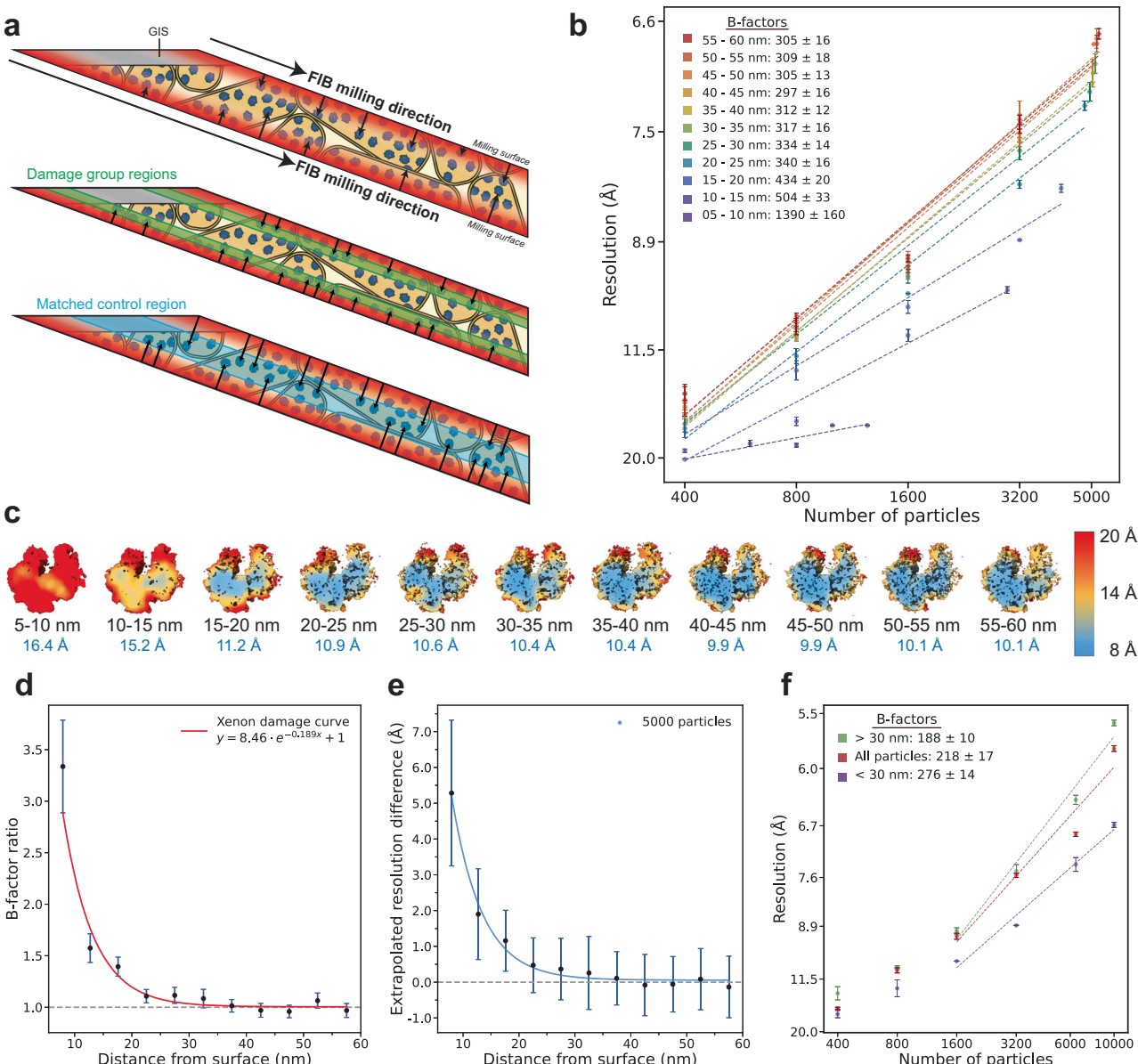

**Fig. 4 | Surface damage analysis of xenon plasma milling using STA. a** Schematic overview of the distance groups and matched controls. The distance of each ribosome to the nearest milling surface was calculated (top), and ribosomes were grouped based on this distance in 5 nm bins (middle). To correct for other factors that affect STA resolution (e.g. local lamella thickness, tilt-series alignment quality), matched controls were created for each distance group by randomly taking the same number of particles from the same tomograms, but further away from the milling surfaces (bottom). **b** B-factor plots for different 5 nm distance groups from the milling surfaces, up to 60 nm. The 0-5 nm group was excluded from B-factor analysis due to an insufficient number of particles. Resolution is plotted on square inverse scale and particle number is plotted on $\log_e$ scale, with resolutions reported as mean ± SE ($n = 3$ technical replicates of the B-factor analysis). **c** Local resolution maps for one of the $n = 3$ ribosome structures obtained during the B-factor analysis in **b** for 800 particles, with the distance group (black) and obtained global resolution (blue) indicated below each map. **d** Plot of the ratio of mean B-factor for each distance group to its corresponding matched control fitted with a three-parameter exponential decay function ($R^2 = 0.89$). Error bars report SE of the B-factor ratio (derived from $n = 3$ technical replicates of the B-factor analysis). **e** Plot of extrapolated difference in resolution (mean ± SE) for 5000 particles between the distance groups and the matched control groups. Exponential decay function fit: $y = 22.4e^{-0.185x} + 0.0568$ ($R^2 = 0.98$). **f** B-factor plots for 10,000 particles randomly selected from the sets of: all particles, particles less than 30 nm from the milling surface, particles greater than 30 nm from the milling surface. Linear fit plotted through the linear section of the data. Resolution is plotted on square inverse scale and particle number is plotted on $\log_e$ scale, with resolutions reported as mean ± SE ($n = 3$ technical replicates of the B-factor analysis). Source data for all plots are provided as a Source Data File.

(Fig. 4d). Studies on the effect of FIB surface damage on biological samples use notably different methodologies to analyse the damage penetration depth, each of which have relative merits[2,19,27–29,33]. Regardless of the technique used, it is important to correct for variations in factors such as local lamella thickness, motion and charging. In this study, we use matched controls for the different depth groups as an internal control, to correct for these per-tomogram

variables as used previously for argon[2]. The way in which different studies define the depth to which damage penetrates also varies, which convolutes the ability to directly compare reported relative damage penetration between ion sources. We therefore suggest that other ion-specific factors such as milling rate, probe size and curtaining propensity should primarily be considered when choosing an ion source for lamella preparation.

**Table 3 | B-factor analysis of xenon PFIB milling surface damage**

| Surface distance groups | Number of particles per distance group | Number of tomograms per distance group | Average lamella thickness (nm)[a] | B-factor (Å²)[b] | Global resolution (Å)[c] |
|---|---|---|---|---|---|
| All particles | 10,000 | 163 | 189 ± 30 | 218 ± 17 | 5.8 |
| <30 nm matched control | 10,000 | 163 | 189 ± 30 | 276 ± 14 | 6.7 |
| >30 nm matched control | 10,000 | 163 | 189 ± 30 | 188 ± 10 | 5.5 |
| 0–5 nm | 252 | 91 | 186 ± 30 | [a] | [a] |
| 0–5 nm matched control | 252 | 91 | 186 ± 30 | [a] | [a] |
| 5–10 nm | 1243 | 152 | 188 ± 30 | 1390 ± 160 | 15.8 |
| 5–10 nm matched control | 1243 | 152 | 188 ± 30 | 417 ± 30 | 9.7 |
| 10–15 nm | 2967 | 162 | 186 ± 29 | 504 ± 33 | 9.9 |
| 10–15 nm matched control | 2967 | 162 | 186 ± 29 | 320 ± 19 | 7.6 |
| 15–20 nm | 4143 | 161 | 189 ± 30 | 434 ± 20 | 8.2 |
| 15–20 nm matched control | 4143 | 161 | 189 ± 30 | 311 ± 15 | 7.1 |
| 20–25 nm | 4795 | 163 | 191 ± 30 | 340 ± 16 | 7.2 |
| 20–25 nm matched control | 4795 | 163 | 191 ± 30 | 307 ± 10 | 6.9 |
| 25–30 nm | 4946 | 162 | 192 ± 31 | 334 ± 14 | 7.2 |
| 25–30 nm matched control | 4946 | 162 | 192 ± 31 | 299 ± 17 | 6.7 |
| 30–35 nm | 5119 | 162 | 192 ± 31 | 317 ± 16 | 7.0 |
| 30–35 nm matched control | 5119 | 162 | 192 ± 31 | 293 ± 19 | 6.7 |
| 35–40 nm | 5048 | 161 | 192 ± 31 | 312 ± 12 | 7.1 |
| 35–40 nm matched control | 5048 | 161 | 192 ± 31 | 308 ± 14 | 6.9 |
| 40–45 nm | 5163 | 162 | 192 ± 32 | 297 ± 16 | 6.9 |
| 40–45 nm matched control | 5163 | 162 | 192 ± 32 | 307 ± 14 | 6.8 |
| 45–50 nm | 5076 | 163 | 193 ± 32 | 305 ± 13 | 6.8 |
| 45–50 nm matched control | 5076 | 163 | 193 ± 32 | 318 ± 15 | 7.2 |
| 50–55 nm | 5170 | 161 | 191 ± 31 | 309 ± 18 | 6.8 |
| 50–55 nm matched control | 5170 | 161 | 191 ± 31 | 291 ± 12 | 6.8 |
| 55–60 nm | 5235 | 162 | 194 ± 30 | 305 ± 16 | 6.8 |
| 55–60 nm matched control | 5235 | 162 | 194 ± 30 | 315 ± 15 | 6.8 |

[a]Average lamella thickness for each particle per distance group.
[b]Reported as mean ± SE.
[c]For one of $n = 3$ repeats.
[d]Too few particles for reliably determining the B-factor.

Methods to reduce the impact of FIB surface damage on STA require further exploration, particularly where the protein of interest is low abundance, or the biological question requires high resolution. Computationally down weighting particles near the milling surfaces has been shown to improve the reconstruction quality for single particle analysis[28], and could be implemented in STA processing pipelines. This would require a highly accurate rapid method to automatically annotate the milling surfaces in tomograms, as manual annotation is low throughput. An alternative is to reduce the accelerating voltage of the ion beam in the final polishing steps, which has precedent in materials science[16,24] and has recently been demonstrated for gallium FIB for biological samples to reduce the penetration depth of surface damage[28]. A practical consideration is the increased probe size at lower voltages, which makes it more challenging to prepare thin lamellae and increases the curtaining propensity[28]. Developing reliable low-kV

polishing protocols for preparing high-quality lamellae will be critical to benefit from this reduction in surface damage.

In this work we describe a high-throughput methodology for preparing cryo-FIB lamellae from high-pressure frozen samples with xenon plasma with currents up to at least 60 nA. The approach allowed a 4.0 Å resolution *E. coli* 70S ribosome to be determined using STA. We observed that the use of such currents led to the formation of an amorphous and striated area on the backside of the lamellae, though B-factor analysis suggests that no significant damage propagates beyond the regions that are visibly affected. Our STA and B-factor calculations determined that 30 kV xenon PFIB milling results in substantial damage up to a depth of 30 nm from the milling surfaces, the effects of which become negligibly small within 30–45 of the surfaces. Overall, this work demonstrates the suitability of xenon PFIB milling for high-resolution in situ structural biology on high-pressure frozen samples.

# Methods

## Sample preparation

*E. coli* C43 (DE3) transformed with plasmids as for the plasmid immunity assay specified by Hogrel et al.[34] were incubated overnight at 37 °C in a shaking incubator in LB supplemented with 100 μg/μl ampicillin, 50 μg/μl spectinomycin and 25 μg/μl tetracycline. The starter culture was used to inoculate a culture at an $OD_{600}$ of 0.1 in antibiotic-supplemented LB, which was grown to an $OD_{600}$ of 0.6 to recover exponential growth before the addition of 0.2% (w/v) D-lactose and 0.2% (w/v) L-arabinose at 16 °C overnight to induce expression of the plasmid system. The bacteria were then resuspended with glycerol in LB to a 10% (v/v) final concentration of glycerol and concentrated by centrifugation at 1000 x *g* immediately before vitrification.

*S. cerevisiae* (Lesaffre commercial baking yeast) was grown in YPD media supplemented with 5% glucose at 30 °C in a shaking incubator to an $OD_{600}$ of 0.6. The yeast sample was concentrated by centrifugation at $500 \times g$ and resuspended with glycerol in YPD to a 10% (v/v) final concentration immediately before vitrification.

## High-pressure freezing

Bacteria were vitrified by high-pressure freezing in a Leica EM Ice (Leica Microsystems), following an adapted version of the 'Waffle' method[23]. 3 mm planchettes (Aluminium HPF carrier type B, Science Services) were polished with fine grit sandpaper up to a final grit of 10,000 and metal polish before coating to remove the concentric surface patterns. All planchettes were coated with soya lecithin solution (1% w/v dissolved in chloroform, Sigma-Aldrich) to aid planchette separation after freezing. The waffle was assembled in the following order: planchette hat flat side up, a 200 mesh Cu grid (Agar Grids 200 Mesh Copper 3.05 mm, Agar Scientific) glow-discharged on both sides, 5-10 μl of bacterial suspension, planchette hat flat side down. Frozen waffles were disassembled under liquid nitrogen. Vitrified grids were clipped into Autogrids with a cut-out notch (Thermo Fisher Scientific) and stored under liquid nitrogen until lamella fabrication.

Yeast samples were vitrified by high-pressure freezing as described above for bacterial samples, except that 6 mm planchettes (Au-plated Cu carrier type B, Leica) were used and 10-20 μl of yeast suspension applied per grid.

## Lamella fabrication

Clipped autogrids were loaded onto an Arctis dual-beam FIB/SEM microscope (Thermo Fisher Scientific). Ice contamination was removed from the front and the back of the grid by FIB imaging with xenon plasma at 15 and/or 60 nA at 30 kV, 38° relative to the grid at the lowest possible magnification (212×), until most of the ice was removed. The front side of the grids was sputter coated with platinum for 120 s at 12 kV and 70 nA, followed by 120 s of coating using the gas injection system (GIS) with trimethyl(methylcyclopentadienyl)platinum(IV), followed by 120 s of sputter coating with platinum using the same conditions as before. From the backside of the grid at −128° stage tilt (where the FIB is 90° relative to the grid), two trenches were milled per lamella site, where all lamella sites were positioned at low magnification for serial milling. Trenches were prepared with a height and width of 30 and 25 μm respectively, using cleaning cross sections at 60 nA, 30 kV. The distance between trenches for a given lamella site was 25 μm. Material below the lamella sites was removed at 45°, 28° and 20° relative to the grid (7°, −10° and −18° stage tilt respectively) using a rectangular pattern with a width of 22 μm and a height between 18 and 22 μm at 15 nA, 30 kV. Patterns were placed right below the GIS layer to cover the bottom trench and underside of the lamella. Multiple patterns were concurrently placed in horizontal rows for milling.

After trench milling, lamella sites were identified in Maps 3.21 (Thermo Fisher Scientific) on an SEM image of the milled grid acquired at eucentric height. Sites were then loaded into AutoTEM Cryo 2.4 (Thermo Fisher Scientific) for automated milling with xenon at 30 kV at

a milling angle of 20° relative to the grid. For each lamella site, eucentric height was refined automatically before user determination of the desired final lamella position and width. Three rough milling steps were performed at decreasing currents of 4.0 nA, 1.0 nA and 0.3 nA as the milling patterns were placed with decreasing distance from the intended lamella position (Table 1). Optionally, a notch was then milled manually at 0.1 nA, before re-defining the final lamella site in AutoTEM Cryo. Lamellae were then polished with 30 pA and 10 pA with a distance between the patterns of 260 nm and 120 nm respectively. The AutoTEM target thickness of 120 nm was empirically determined to result in ~200 nm thin lamellae using xenon polishing.

## Cryo-electron tomography

Data were collected with Tomo5 software (Thermo Fisher scientific) on a Titan Krios G4 (Thermo Fisher Scientific) TEM with a Falcon 4i camera and a Selectris X energy filter. Low-dose tiled overviews were collected at a 20 degree angle to correct for the lamella pre-tilt and tilt series were collected dose-symmetrically[6] in electron counting mode with the following parameters: magnification of 64,000 × (calibrated pixel size of 1.90 Å/px), starting tilt angle of 20°, tilt range of ± 51° with 3° increment. A dose per tilt image of 5 e−/Å$^2$ was used for *E. coli* and 3.5 e−/Å$^2$. for *S. cerevisiae*. The target defocus ranged between 3 and 5 μm across tilt series, in 0.5 μm increments. Tomo5 was used to collect tilt-series in EER format using multi-shot beam shift acquisition[7,8].

## Tilt series processing

Tilt images with low contrast or objects moving into the field of view (e.g. grid bars or ice contamination) were manually excluded using a custom script. Warp[9] (version 1.0.9) was used for gain correction, CTF estimation, motion correction using a frame group size of 10 and tilt series stack generation for all tilt-series. Tomogram volumes were reconstructed with AreTomo[35] (version 1.2.5) using simultaneous algebraic reconstruction technique (SART) at a binning factor of 8 and a Z-height of 386 pixels (15.2 Å/px) and flipped with IMOD[36] (version 4.11.1). For filtering the tomograms we used either a low-pass filtered using EMAN2[37] which thresholds the frequencies, CTF deconvolution using Isonet which uses a Wiener-like filter to boost contrast lost due to the contrast transfer function[38,39], or machine learning based missing wedge compensation using Isonet[38]. An Iso-Net model was trained on 13 selected *E. coli* tomograms, using a cube size of 32 and crop size of 128 without any masking, and applied to all the tomograms.

## Ribosome particle picking

For training a crYOLO[40] model for particle picking, all the ribosome particles were manually picked every ~25 slices from 8 tomograms (3337 total particles). These co-ordinates were used to train four crYOLO models on (1) unfiltered tomograms, (2) low-pass filtered tomograms, (3) CTF deconvolved tomograms (as implemented in IsoNet), and (4) IsoNet filtered tomograms. Predictions of ribosome co-ordinates on all tomograms resulted in higher mean confidence scores using the model trained on IsoNet filtered tomograms (Supplementary Fig. 4), therefore these particle position predictions were used for further processing. This gave 224,823 putative ribosome particles picked above the chosen confidence threshold value of 0.45.

## Sub volume averaging

All 224,823 predicted ribosomes were extracted at a binning factor of 4 (7.6 Å/px) with a cube size of 64$^3$ voxels using Warp and imported into RELION 4.0[10]. A subset of 5415 particle volumes from 6 manually selected tomograms was 3D classified (using the non-aligned average of this subset as an initial reference), and the most ribosome-like class was used as a reference for 3D refinement of the subset to give a ribosome structure with resolution 16.2 Å. This subset-derived

structure was lowpass-filtered to 60 Å and used as a reference for 3D refinement in RELION of the whole dataset, followed by two rounds of 3D classification to identify 95,679 ribosomes. These particles were re-extracted in Warp at a binning factor of 2 (3.8 Å/px) and then 3D refined in RELION to give a ribosome structure with resolution of 7.6 Å.

106 out of 286 tomograms were excluded from further processing using the following criteria: 1) the contrast of the tomogram did not allow for a confident annotation of the milling surfaces, 2) ribosomes could not be clearly identified in the tomogram, or 3) the tomogram contained substantial redeposition or other surface contamination. Particles positions were imported into M and an initial round of M was run with no refinement parameters selected to verify that the pro-gramme could run successfully. An additional 13 tomograms were excluded during processing in M due to software errors, resulting in a total of 82,377 particles from 167 tomograms being used for the final reconstruction in M. The following parameters were then applied incrementally over six subsequent rounds of refinement: (1) particle poses, (2) 3×3 image warp grid, (3) 3×3×2×10 volume warp grid, (4) 3×3×2×10 volume warp grid, (5) temporal poses = 3, (6) defocus optimisation with grid search in 1st iteration. This resulted in a ribo-some structure with a resolution of 4.0 Å, based on a Fourier-shell correlation using the 0.143 criterion. Panels containing density maps and structures were prepared with ChimeraX[41,42].

### Lamella backside damage analysis

Lamella overviews were created by stitching the search images recor-ded by Tomo5 using the FIJI[43] plugin "Grid/Collection stitching[44]. The positions where tilt-series were acquired were manually found using the first tilt-image recorded for each tilt-series, the central position of each search image in the stitched lamella overview and screenshots of the position placement in Tomo5. Using the measure tool in FIJI, the distance to the boundary where the biological contrast disappears on the backside of the lamella were then measured. This value was then modified per-particle by adding the displacement of the particle X coordinate from the centre of the tomogram to this distance, which was added as a separate column to the.star file. Separate.star files were prepared in 1 μm distance (Table 2) groups using Starparser (https://github.com/sami-chaaban/starparser) and subsets of particles were independently refined in RELION 4.0 using the Relion bfactor_plot script (https://github.com/3dem/relion/blob/master/scripts/bfactor_plot.py). B-factor calculations were performed as described below. The average length of the amorphous regions was determined for each lamella by manually segmenting them in FIJI[43], calculating the area, and dividing it by the length of the segmented region perpendicular to the milling direction.

### FIB surface damage depth analysis

Damage depth analysis was performed as described previously[2]. Briefly, lamella surfaces were manually annotated every ~100 slices and interpolated using a custom script (https://github.com/rosalindfranklininstitute/RiboDist). The script determines the tomo-gram thickness and the shortest distance for each ribosome to the interpolated boundary models and are saved in a STAR file. This is also the method used in this study (both for *E. coli* and *S. cerevisiae* data-sets) to estimate lamellae thicknesses. Separate STAR files were pre-pared using Starparser for all ribosomes 0 to 60 nm from the boundary models in 5 nm distance groups (Table 3) and as controls, ribosomes randomly selected from the same tomograms that are further away from the boundaries. If insufficient particles were present in a matched control compared to the corresponding distance group, particles were removed at random from the distance group to obtain the same number of particles as the matched control. B-factors were determined as described below, using resolutions derived from a minimum of 400 particles. The 0-5 nm distance group was excluded from B-factor analysis, as too few particles were present.

### B-factor calculations and error measurements

For each group of data, B-factor analysis scripts were run three times, such that for each particle number group there are $n = 3$ resolution data points. The implementation of the B-factor script means that each particle subset is drawn randomly from the whole set of particles of that group. The mean and standard error of the resolution for a given particle number were calculated and the error propagated according to Eq. (1) for B-factor plots.

$$y = \frac{1}{Resolution^2} \tag{1}$$

Ordinary least squares regression linear fits were performed on all data points of ln(*particle number*) vs $1/resolution^2$ per B-factor group (unless a more limited range specified for linear fitting). B-factors were calculated with Eq. (2), and standard errors of B-factors were calcu-lated by propagation of the standard error of the slope fit according to Eq. (2).

$$B - factor = \frac{2}{slope} \tag{2}$$

B-factor ratios were calculated by Eq. (3) and the standard error of the ratio propagated accordingly.

$$B - factor\ ratio = \frac{B - factor_{depth\ group}}{B - factor_{matched\ control}} \tag{3}$$

Predicted resolutions for 5000 particles were extrapolated from the linear fit of the B-factor plots for both depth and matched control groups. The error in predicted $y$ of each group (in $1/resolution^2$) was calculated as $\sigma_y = \sqrt{(\sigma_m \cdot x)^2 + \sigma_c^2}$, then propagated to the error in resolution measurement according to $y = \frac{1}{Resolution^2}$. The resolution difference was then calculated between the two groups per depth according to Eq. (4), and error in the resolution difference was pro-pagated accordingly.

$$\Delta Extrapolated\ resolution = Extrapolated\ resolution_{depth\ group}$$
$$- Extrapolated\ resolution_{matched\ control} \tag{4}$$

Three parameter exponential curve fitting was performed weighted for the standard error in either the B-factor ratio or the extrapolated resolution difference between depth groups and its matched control.

### Reporting summary

Further information on research design is available in the Nature Portfolio Reporting Summary linked to this article.

## Data availability

The data that support this study are available from the corresponding authors upon request. Subtomogram averages of ribosomes obtained in this study were deposited in the Electron Microscopy Data Bank (EMDB): full reconstruction (EMD-50675), the reconstructions from the 1 μm distance groups from the backside damage (EMD-50699, EMD-50700, EMD-50701, EMD-50702, EMD-50703), 5-nm distance groups for the PFIB surface damage (EMD-50676, EMD-50678, EMD-50679, EMD-50680, EMD-50681, EMD-50682, EMD-50683, EMD-50684, EMD-50685, EMD-50686, EMD-50687) and the matched controls (EMD-50688, EMD-50689, EMD-50690, EMD-50691, EMD-50692, EMD-50693, EMD-50694, EMD-50695, EMD-50696, EMD-50697, EMD-50698), and the 10,000 particle sets for determination of ion beam damage impact (EMD-52177,

EMD-52178, EMD-52179). The raw microscope data (frames and meta-data files) used for subtomogram averaging and associated processed data are available on the EMPIAR data server under accession code EMPIAR-12413. PDB model 6ORE[45] used as a representative ribosome model for structure fitting. Source data are provided with this paper.

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

## Acknowledgements

We thank Thomas Glen and Matthew Case for microscope support, Neville B.-y.Yee for help with the RiboDist software and Sheera

Abdulla for wet lab support. We thank Gaëlle Hogrel and Malcolm F. White for providing the plasmids. We would also like to thank Laura Spagnolo and Malcolm F. White for guidance with bacterial work. This work was supported by the Wellcome Trust through the Electrifying Life Science project (220526/Z/20/Z to J.H.N.) and a Wellcome Career Development Award (225902/Z/22/Z to M.G.). The Rosalind Franklin Institute is funded by UK Research and Innovation through the Engineering and Physical Sciences Research Council (EPSRC).

## Author contributions

C.B., H.W., M.D., and M.G. conceived the study. H.W. prepared cryo samples. H.W. and C.B. prepared PFIB lamellae, collected TEM data, performed STA and damage analyses. C. B., H.W., M.D. and M.G. wrote the manuscript and C.B. and H.W. prepared figures. M.D. and M.G. supervised the project. C.B., H.W., J.H.N., M.D., and M.G reviewed the manuscript and the data.

## Competing interests

The authors declare no competing interests.
