## [Transparent Peer Review file · Nature Communications]

Xenon plasma focused ion beam lamella fabrication on high-pressure frozen specimens for structural cell biology

Corresponding Author: Dr Michael Grange

Version 1:

Reviewer comments:

Reviewer #1

(Remarks to the Author)

In this work, Berger et al. describe the adaptation of the “waffle method” FIB milling workflow (Kelley et al., 2022) to a plasma FIB instrument improving the workflow significantly at several steps. They show a high-resolution structure obtained from high pressure frozen samples and further characterise the damaging of the sample by the high-current milling procedure. This complements previous work investigating the damage layers caused by FIB milling with Argon and Gallium ion sources.

This work will be very useful to the field as plasma FIB milling instruments are becoming more and more common for in situ structural biology sample preparation. The here described workflow has great potential to become a widely used method to access previously inaccessible samples for high-resolution cryoET imaging in a high-throughput manner. In fact, we started to use the workflow ourselves and already benefitted from the work done here.

We have one major point and suggest several minor points and some clarifications that could make the manuscript more accessible to readers.

Major point:

The results of this manuscript are based on one single milling session on one grid. We would like to encourage the authors to repeat their milling strategy on another grid (maybe different biological sample) to demonstrate the robustness of their workflow. While no high-resolution subtomogram average is needed, the quality of lamellae on the second grid could be analyzed by 2D cryoEM imaging.

Minor points that the authors may want to consider (in order of occurrence in the manuscript):

Abstract

1. Consider splitting up the sentence in lines 22-25 for clarity.
2. Typo in line 29: “Ours results outlines”

Introduction

1. The recent advancements in serial lift-out procedures should be mentioned briefly with advantages and disadvantages as this will be the other alternative for higher-throughput lamella preparation that can be used on samples too thick for the waffle method. (Currently, serial lift-out is only alluded to as “novel milling strategies” in line 56.)
2. Since voltage was brought up in the previous paragraph, line 72 should specify that 60 nA at 30 kV was used for clarity.

Results

1. Is there something more you could say about the type of ice contamination in lines 87-88? Maybe a size range or possible causes?
2. It should be mentioned that the cleaning step is done before coating. Consider adding a comment on ice contamination accumulating on the grid bars, presumably due to charging effects?

3. In line 96, you write "currents up to 60 nA". I did not see any other reference to trench milling currents other than 60 nA. Did a lower trench milling current possibly show a reduced amorphous area at the back of lamellae?
4. The waffle method publication (Kelley et al., 2022) emphasized that the removal of material below the lamella at increasingly shallower angles should always be done with a pattern partially overlapping with the GIS layer for protection. Was this also done here?
5. Fig. 1f in line 99 should be Fig. 1d.
6. Why was the notch milled after rough milling rather than before rough milling as published in Kelley et al., 2022?
7. Figure 1:
 - a. In panel c), consider showing the same part of the image before and after cleaning for better comparison. Additionally, the legend says, "top left" and "bottom right" which might refer to an earlier iteration of the figure with diagonal boundary?
 - b. In the figure legend, consider saying "ice contamination removal" instead of "ice removal" to avoid confusion (line 108).
 - c. What does "example milled sites" in line 114 mean? Is one of them the site shown in panels h-s?
 - d. Consider numbering the trench milling steps in panel d to make it clear which one the 4th step is.
 - e. Check panel letters after "for stress-relief" in line 123 and after "Scalebars" in line 124.
 - f. Consider making all scalebars in panels h-s 10 μm to make changes in magnification more obvious.
8. Table 1:
 - a. I think "FIB angle relative to grid" should be 38 degrees for Ice cleaning steps.
 - b. Pattern overlap [%] should be defined unambiguously for applications outside of autoTEM.
9. It is stated that 20 out of 24 prepared sites were used for milling. What were the reasons to exclude the remaining 4 sites?
10. Supplementary Figure 1:
 - a. In line 486, "tomograms suitable" should be "lamellae suitable".
 - b. Why is the most left lamella considered suitable, but 0 tilt series have been collected?
 - c. Consider adding the number of tilt series used for the final average from each lamella.
 - d. It seems that low mag images and medium mag images were collected at different stage tilts when looking at the lamella in panel e. Please add the stage tilt of the images in the figure legend.
 - e. Consider showing where tilt series were collected in panels b-e and the thickness estimates from the tomograms.
11. The mean local thickness of 200 nm is much larger than the target thickness of 120 nm according to table 1. Please consider adding a comment on that similar to Berger et al., 2023. What is the reason for the systematic offset compared to Ga ions, is it the probe size?
12. Since the front of lamellae is considerably thinner than the back of lamellae, did you have a specific reason not to try using overtilt as it was used in Kelley et al., 2022 and is readily implemented in autoTEM? Is this thickness gradient more or less pronounced compared to Argon or Gallium ion milling in your experience?
13. The particle number in line 136 is 1000 less than the number mentioned in Sup. Fig. 3 or in the methods section.
14. Supplementary Figure 3:
 - a. Consider specifying the Relion version in the figure since it has an impact on the algorithms used.
 - b. Consider showing discarded classes in 3D classification step. False positive picks? Ribosomal subunits?
15. Since a large part of the structure was limited by the Nyquist sampling but data was collected in EER format, why did you not use upsampling to further explore the resolution limit of the data?
16. Figure 2:
 - a. Panel a could be a bit more clear and less verbose. Maybe consider using a bar plot for the pFIB/SEM panel as well to show how many lamellae are retained. Since the workflow was described in figure 1, the left box describing the step could be omitted. You could also consider using a lightly coloured background for the 3 panels to reduce the concentration of dashed lines between the pFIB/SEM and TEM panels.
17. The striated layer at the back of the lamella is very striking. Do you have a theory what it could consist of? Did you observe it in weaker forms in other contexts (different ion species)?
18. Do you observe the amorphous area on all of your lamellae? Could you add a plot showing the size distribution of that area depending on lamella?
19. Supplementary Figure 7:
 - a. The sidewall damage should also be indicated on the bulk material outside the lamella site (left and right blocks).
 - b. Consider also indicating a presumably smaller damage layer from the 15 nA undercut that is subsequently removed throughout the rough milling steps.
20. Supplementary Figure 5:

a. Why was a different colour map used to display local resolution compared to Figure 2g?

21. In line 219, is “by variability” supposed to be “by a variety”?

22. The sentence in lines 218-221 could be clearer. (It is much clearer in the figure 4 legend.)

23. Figure 4:

a. In panel a, what I assume to be the GIS layer (grey leading edge) should be labelled. Consider also to adjust the lamella geometry to be more similar to Fig. 1g bottom (GIS layer and back trench not perpendicular to FIB).

b. In panel c, why is the colour map for local resolution different from figure 2g?

c. In panels d and e, how were the curves fitted?

Discussion:

1. As mentioned above, consider speculating on the mechanism of the cleaning procedure and if it would work with other ion species.

2. In line 276 and 330, a “~2 μm damaged area” is mentioned. In the results section it was referred to as “approximately 1 μm ” in line 171 and “0.5 to 1.5 μm ” in line 200. As mentioned above, I think a plot of the size distribution of these areas would be helpful.

3. As mentioned in the Introduction section, a small discussion of the advantages and disadvantages compared to “high-throughput” lift-out would help to put this advancement of the waffle method in context.

Methods

1. Please add the version number for Maps in line 376.

2. Consider adding an image or the dimensions of the line pattern used for notch milling.

3. I would like to encourage you to publish the autoTEM templates used for the shown lamellae with this manuscript.

4. In line 389, it is stated that the overviews were collected at 20 degrees tilt. Is this also the start tilt angle for the dose-symmetric tilt series acquisition?

5. In line 399, a custom script for excluding tilts is mentioned. Does this script simply remove the corresponding blocks from the mdoc file, or does it have additional functionality?

6. The benchmarking of filters for particle picking is great and should be mentioned in the main text to make sure readers will come to the methods section to learn more.

7. In line 423, does box size refer to a 64x64 image or a 64x64x64 volume?

8. In line 425, does “using the average” refer to the output of a 3D auto refine run or an average without alignment?

9. In lines 426-427, was the reference lowpass filtered or used at 16.2 \AA ?

10. In lines 432-435, do you have the data on how many tomograms were excluded due to each reason?

11. In line 437, do you have a guess as to what caused the software problems with these 13 tomograms?

12. How was local resolution determined?

13. In lines 448-449, were the positions of tilt series found by eye or by cross-correlation/image registration?

Supplementary Movies

1. Consider adding the total reconstructed z-height of the tomograms shown in Supplementary movie 2 to the legend.

Data availability

1. I would like to encourage you to also deposit the raw data as well as the lamella overviews and grid map.

Co-reviewed by Fabian Eisenstein/Martin Pilhofer

Reviewer #2

(Remarks to the Author)

Reviewer #3

(Remarks to the Author)

In this manuscript the authors describe application of Xenon plasma FIB milling in cryogenic conditions to generate lamellae from high-pressure frozen cells using an Arctis cryo-plasma FIB-SEM, of which they have an early commercial release. Generating thin lamellae is essential for in situ structural biology using cryo-EM and cryo-ET. Current approaches use a Gallium FIB which has the limitation that the relatively low current density makes it impractically slow when milling thick samples such as high-pressure frozen cells, limiting interrogation of tissue samples. In this paper the authors describe using a Xenon plasma source, which has the dual benefit of a higher current density, allowing higher currents to be used for milling, and a higher atomic mass, increasing the sputtering yield: effectively making milling faster.

To demonstrate this, the authors use high-pressure frozen E.coli samples. They describe some of the practical strategies that they needed to use such as removing ice contamination with a low magnification PFIB imaging and the angles and different currents used which will be of use to others wishing to try this approach. They show that they are able to generate a 4Å reconstruction of the ribosome using subtomogram averaging, and use subtomogram averaging of ribosomes at different depths from the lamella surface to measure damage from milling. The results from this study are aligned with expectations from simulations and from decades of work in the material sciences. This study will be useful to the field as a practical guide for the direct application to cryogenically frozen biological samples. The manuscript would benefit from more information about milling times.

Comments below:

-In Figure 3 the authors note an amorphous area on the backside of the lamella that appears vitreous and without features. The authors propose that this is due to damage from the initial milling at 60nA. Since 1) there is a distinct barrier between this layer and the rest of the lamella, 2) it only appears on the backside and 3) there is an alternating pattern of high and low electron dense material can the authors rule out that this is redeposited material rather than damage to the extant lamella? The authors should consider alternate causes of this layer and not claim that it is the result of 60 nA milling unless they have a contrast with different currents.

-For Figure 4 I appreciate the use of matched controls and attempts to measure the damage curve rather than define a specific cut off for damaged/undamaged. This more rigorous step was not performed in their previous analysis of Argon FIB milling damage, so a direct comparison is difficult. The authors should be more cautious in their conclusion that Xenon and Argon have similar damaging propensities.

-Since one of the major arguments for using Xenon FIB milling is the potential time improvement, the authors should include a time estimate for each of the milling steps/ overall time to get that number of lamellae (and potentially contrast with Argon and Gallium). This could be included in Figure 2A. This result would be of great interest to the field.

-What is the function of the curve fit in figure 4e and f? and what is the accuracy of this fit?

Version 2:

Reviewer comments:

Reviewer #1

(Remarks to the Author)

The authors added a considerable amount of data from an additional biological sample to the manuscript and demonstrated reproducibility of their presented workflow.

All our concerns were addressed and especially the comparison between the damage layer using 4 nA and 60 nA added an important piece to the data.

The presentation is more clear and contains additional statistics supporting the results.

Therefore, we recommend the publication of this improved manuscripts pending some minor text revisions:

*Line 143: "lalla" typo

*Lines 302 and 314: duplicated paragraph

*Lines 417-418: disjointed sentence

Reviewer #2

(Remarks to the Author)

Reviewer #3

(Remarks to the Author)

Overall I think that the paper is reporting a useful proof of principle: that lamella milled with Xenon can be used to generate subtomogram averages, and one suggested strategy to do so. In continuance of my earlier points, I think that any quantitative claims need to be backed up with statistics or de-emphasized.

Thank you to the authors for now ambiguating about the cause of the amorphous layer on the backside of the lamella. I would encourage the authors to limit their emphasis on this layer. I would focus on the absence of evidence of devitrification,

because this one of the concerns about increasing current, the ability to access higher currents is a key advantage of Xenon. I maintain that the observed pattern is inconsistent with damage to the extant lamella and consistent with redeposition during milling. Whether there is damage closer to that edge of the lamella (i.e.: at <1 micron) was not assessed. The striations would be an expected result of redeposition when milling through the GIS layer and the cellular material resulting from the back milling strategy.

How the authors defined the depth of damage remains unclear. I did not find a statistical or quantitative justification for making this call. In figure 4d and corresponding sup figs, the B-factor ratios are close to 1 at ~30 nm. However, there is some variation in these exact B-factor ratios (not unexpected) and, consequently, it is unclear what the built in error in their measurement is. (Perhaps this could be estimated from the variation in the "matched control" sets?) It is unclear how they conclude a depth of 45 nm from these data. Are the authors using the point at which the fitted curve crosses $y=1$? In the figure 4f visually the fitted Xenon curve does not cross 1 in the shown range and the Argon curve crosses at ~42nm, if you solve the listed equations for $y=1$ you get different values: Figure 4f: Argon $y=1$ at 49nm, Xenon $y=1$ is undefined. Are the authors sure that the equations listed matches the curves shown? From the equation in 4d, Xenon $y=1$ at 50nm. The curve fits have a reported r^2 of 1, despite one of the values clearly not fitting the curve in figure 4d, and the fitted equation varies between experiments. This suggests to me that this study is underpowered to define the depth of damage beyond a statement that there is appreciable damage within 30nm of the lamella surface.

The fitted curve equations have too many significant figures for the data shown and the text in the equations shown are too small to read all the terms.

The figure in 4e allows them to say only that the group of particles <45nm from the surface and those >45nm from the surface appear to have different B-factors, not that the damage extends to 45nm. The data points do not appear to be in a linear range.

Lines 302 to 325 in the discussion contain repeated content.

It is not relevant to the conclusions in paper but may be for future biologists who may want to use the data after deposition in EMPIAR: the images of yeast look unusual, what strain was used?

Version 3:

Reviewer comments:

Reviewer #3

(Remarks to the Author)

I thank the authors for adding error bars to their resolution and B-factor ratios and for attempting to limit their B-factor estimation to a more linear range. The additional information and discussion of caveats will allow readers to make a more informed interpretation of these data with the inherent uncertainty.

I think this work will be of interest to the field as we all try to work out the best way of preparing lamellae for different sample types.

REVIEWER COMMENTS

Reviewer #1 (Remarks to the Author):

In this work, Berger et al. describe the adaptation of the “waffle method” FIB milling workflow (Kelley et al., 2022) to a plasma FIB instrument improving the workflow significantly at several steps. They show a high-resolution structure obtained from high pressure frozen samples and further characterise the damaging of the sample by the high-current milling procedure. This complements previous work investigating the damage layers caused by FIB milling with Argon and Gallium ion sources.

This work will be very useful to the field as plasma FIB milling instruments are becoming more and more common for in situ structural biology sample preparation. The here described workflow has great potential to become a widely used method to access previously inaccessible samples for high-resolution cryoET imaging in a high-throughput manner. In fact, we started to use the workflow ourselves and already benefitted from the work done here.

We thank the reviewer for thoroughly reviewing the manuscript and for all the helpful comments and suggestions. We are glad to hear that this reviewer has already been able to benefit from the workflow described in our manuscript.

We have one major point and suggest several minor points and some clarifications that could make the manuscript more accessible to readers.

Major point:

The results of this manuscript are based on one single milling session on one grid. We would like to encourage the authors to repeat their milling strategy on another grid (maybe different biological sample) to demonstrate the robustness of their workflow. While no high-resolution subtomogram average is needed, the quality of lamellae on the second grid could be analyzed by 2D cryoEM imaging.

We repeated our milling protocol in a further two sessions (each milled in a 1 day session) on high-pressure frozen yeast samples to demonstrate its reproducibility and analysed tomogram thickness from the collected datasets. We have updated Figure 2a,b to include this, and have also included an additional supplementary figure (Supplementary Figure 2) with the atlas overviews and representative images of lamellae and tomograms.

Minor points that the authors may want to consider (in order of occurrence in the manuscript):

Abstract

1. Consider splitting up the sentence in lines 22-25 for clarity.

We divided the sentence as suggested.

2. Typo in line 29: “Ours results outlines”

We corrected the typo.

Introduction

1. The recent advancements in serial lift-out procedures should be mentioned briefly with advantages and disadvantages as this will be the other alternative for higher-throughput lamella preparation that can be used on samples too thick for the waffle method. (Currently, serial lift-out is only alluded to as “novel milling strategies” in line 56.)

We changed this sentence in the introduction to explicitly mention serial lift out and the “waffle” method., and added a brief discussion on the relative advantages of serial lift-out compared to waffle in the first paragraph of the discussion.

2. Since voltage was brought up in the previous paragraph, line 72 should specify that 60 nA at 30 kV was used for clarity.

We added the voltage.

Results

1. Is there something more you could say about the type of ice contamination in lines 87-88? Maybe a size range or possible causes?

We added a rough size range for the ice contamination, and now mention in the discussion the HPF is as a likely cause of the ice contamination.

2. It should be mentioned that the cleaning step is done before coating. Consider adding a comment on ice contamination accumulating on the grid bars, presumably due to charging effects?

We clarified that the ice removal step is done before platinum sputtering and GIS deposition. We agree that removing ice contamination from the grid bars appears to be more difficult, but have not done any systemic

measurements over multiple samples to test this hypothesis. We therefore would prefer not to comment on it in the manuscript.

3. In line 96, you write “currents up to 60 nA”. I did not see any other reference to trench milling currents other than 60 nA. Did a lower trench milling current possibly show a reduced amorphous area at the back of lamellae? We changed this sentence to ‘xenon at 60nA’. We included a comparison of perpendicular trench milling at 4 nA and 60 nA (supplementary Figure 7), and 4 nA perpendicular trench milling does indeed significantly reduce the length of the amorphous region at the back of the lamellae.

4. The waffle method publication (Kelley et al., 2022) emphasized that the removal of material below the lamella at increasingly shallower angles should always be done with a pattern partially overlapping with the GIS layer for protection. Was this also done here?

In our experience, overlapping the patterns with the GIS layer for the undercuts is not critical. We now specify in the materials and methods section that patterns are placed right below the GIS layer.

5. Fig. 1f in line 99 should be Fig. 1d.

We corrected this in the manuscript.

6. Why was the notch milled after rough milling rather than before rough milling as published in Kelley et al., 2022?

We did not observe any sample deformation during rough milling steps of the lamella area. We therefore reasoned that the notch can be prepared later. We consider the notch milling to be optional, which we have updated the manuscript to reflect. We note that the notch milling step was omitted in the preparation of lamellae from yeast presented here.

7. Figure 1:

a. In panel c), consider showing the same part of the image before and after cleaning for better comparison. Additionally, the legend says, “top left” and “bottom right” which might refer to an earlier iteration of the figure with diagonal boundary?

We changed panel c to show the same field of view before and after ice removal, and also changed the images to show a clearer example. We corrected the figure caption, which was indeed referring to a previous iteration of the figure panel.

b. In the figure legend, consider saying “ice contamination removal” instead of “ice removal” to avoid confusion (line 108).

We changed the manuscript to use ice contamination removal instead of just ice removal.

c. What does “example milled sites” in line 114 mean? Is one of them the site shown in panels h-s?

Figure 1 is composed from images from multiple different experiments with images that demonstrate the protocol well. The lamella shown in panel h-s was prepared on a different grid than shown in b, e and f.

d. Consider numbering the trench milling steps in panel d to make it clear which one the 4th step is.

We numbered the milling steps in panel d to match the numbering used in Table 1.

e. Check panel letters after “for stress-relief” in line 123 and after “Scalebars” in line 124.

We corrected the panel letters.

f. Consider making all scalebars in panels h-s 10 μm to make changes in magnification more obvious.

We changed all the scalebars to be 10 μm in length.

8. Table 1:

a. I think “FIB angle relative to grid” should be 38 degrees for Ice cleaning steps.

We corrected the listed angles in the table.

b. Pattern overlap [%] should be defined unambiguously for applications outside of autoTEM.

We changed Table 1 to show the absolute pattern height in addition to the pattern overlap %. We also changed the “distance between patterns” column to “offset from lamella” as we used previously (Berger, Dumoux and Glen et. al. 2023). This is because AutoTEM applies an additional offset between the “target thickness” and the distance between the patterns, making our previous values for pattern distance incorrect. We measured the approximate pattern distance for the final milling step and added it to the table.

9. It is stated that 20 out of 24 prepared sites were used for milling. What were the reasons to exclude the remaining 4 sites?

As it only takes ~3 minutes for the first trench milling step per site, it is generally advisable to prepare a few more trench milled sites than required, and exclude any sites where the front grid surfaces are rougher compared to the others. We added a sentence to the figure caption of Fig. 1e to indicate this to the reader, and added to the main text that we select sites for their surface smoothness before proceeding to automated milling.

10. Supplementary Figure 1:

a. In line 486, “tomograms suitable” should be “lamellae suitable”.

Correct, we changed this in the manuscript.

b. Why is the most left lamella considered suitable, but 0 tilt series have been collected?

Tilt series were queued for acquisition on this lamella, but could not be collected as we were limited by the length of our TEM session.

c. Consider adding the number of tilt series used for the final average from each lamella.

Added as suggested.

d. It seems that low mag images and medium mag images were collected at different stage tilts when looking at the lamella in panel e. Please add the stage tilt of the images in the figure legend.

The atlas and lamella overview images were indeed collected at different stage tilts (0 and +20° respectively). We have added this to the figure caption.

e. Consider showing where tilt series were collected in panels b-e and the thickness estimates from the tomograms.

We added a new panel (f) where we show the approximate position where tilt-series were acquired with coloured squares, and the measured thickness of each site. We prefer not to overlay this information for each lamella, as it would make it difficult to see the lamellae themselves. The measured thicknesses are linked to the lamella from which they were acquired in the source data file.

11. The mean local thickness of 200 nm is much larger than the target thickness of 120 nm according to table 1. Please consider adding a comment on that similar to Berger *et al.*, 2023. What is the reason for the systematic offset compared to Ga ions, is it the probe size?

In our experience, the AutoTEM “target thickness” does not correspond well to the thickness of the resulting lamellae with plasma ion sources. As the AutoTEM software (Thermo Fisher Scientific) is not open source code, we are not completely sure how AutoTEM defines “target thickness”. We noticed an offset between the distance of the placed patterns and the “target thickness”, presumably to account for the probe size. We added the measured value of this offset to table 1. Beside the distance between the milling patterns, other factors such as milling time, current, time since the apertures have been last replaced and ion species also impact this and AutoTEM is not transparent on whether these factors are taken into consideration for the target thickness. We added a sentence to the materials and methods section similar to our previous comment in Berger, Dumoux and Glen *et al.* 2023: “The AutoTEM target thickness of 120 nm was empirically determined to result in ~200 nm thin lamellae using xenon polishing”.

12. Since the front of lamellae is considerably thinner than the back of lamellae, did you have a specific reason not to try using overtilt as it was used in Kelley *et al.*, 2022 and is readily implemented in autoTEM?

We agree that using under/overtilt is a good strategy to help reduce the thickness gradient (Schaffer *et al.* 2017). In our limited experience in testing the automated implementation of using under/overtilt, the pattern placement does not appear to be very accurate when tilting, which is why we generally don't use it. We agree that it may be worthwhile to further experiment with this.

Is this thickness gradient more or less pronounced compared to Argon or Gallium ion milling in your experience? in both this study using xenon and in Berger, Dumoux and Glen *et al.* 2023 using argon we observed variability between lamellae in the slope of the thickness gradient, which is also described for Gallium (Tuijtel *et al.* 2024, Fig. S1). We agree that there could be differences in the slope of the thickness gradient between ion sources, but based on our day-to-day experience we cannot see an obvious difference. As other factors such as milling time, pattern distance and current may also affect this, a systematic study directly comparing different ion sources using the same samples and milling methodology with a large enough sample size may be required to show any differences.

13. The particle number in line 136 is 1000 less than the number mentioned in Sup. Fig. 3 or in the methods section.

We corrected the particle number in line 136.

14. Supplementary Figure 3:

a. Consider specifying the Relion version in the figure since it has an impact on the algorithms used.

We have specified this in the figure.

b. Consider showing discarded classes in 3D classification step. False positive picks? Ribosomal subunits?

We have updated the figure to show the discarded classes, which appear to be either false positive picks or contain significant artefacts. None of the identified classes appear to be 50S/30S subunits.

15. Since a large part of the structure was limited by the Nyquist sampling but data was collected in EER format, why did you not use upsampling to further explore the resolution limit of the data?

We agree that this may have resulted in a higher resolution STA average. Because in this study we are not interested in the biology of the *E. coli* ribosome and in the interest of time, we decided not to pursue this option.

16. Figure 2:

a. Panel a could be a bit more clear and less verbose. Maybe consider using a bar plot for the pFIB/SEM panel as well to show how many lamellae are retained. Since the workflow was described in figure 1, the left box describing the step could be omitted. You could also consider using a lightly coloured background for the 3 panels to reduce the concentration of dashed lines between the pFIB/SEM and TEM panels.

We changed panel a following your suggestions and also added the data for the new yeast datasets.

17. The striated layer at the back of the lamella is very striking. Do you have a theory what it could consist of? Did you observe it in weaker forms in other contexts (different ion species)?

We are unsure what causes the striated patterns on the back of the lamellae. We did observe variability in its exact appearance, such as visible in the new Supplementary Figure 7, where it's less pronounced.

18. Do you observe the amorphous area on all of your lamellae? Could you add a plot showing the size distribution of that area depending on lamella?

We added a new figure (Supplementary Figure 7) where we show the distribution of the length of the amorphous layer between lamellae for the bacterial dataset, and a comparison between lamellae with trenches milled at 4 nA and 60 nA for yeast (60 nA yeast dataset = dataset 3 of figure 2a).

19. Supplementary Figure 7:

a. The sidewall damage should also be indicated on the bulk material outside the lamella site (left and right blocks).

b. Consider also indicating a presumably smaller damage layer from the 15 nA undercut that is subsequently removed throughout the rough milling steps.

We have updated the figure to incorporate these suggestions.

20. Supplementary Figure 5:

a. Why was a different colour map used to display local resolution compared to Figure 2g?

We used a different colour map for the local resolutions in both Supp figure 5 and Fig 4c because the local resolutions are substantially lower than in Fig 2g (3.8Å-6.8Å vs 8-20Å), and using the same colour range could give the false impression that they are similar.

21. In line 219, is "by variability" supposed to be "by a variety"?

We corrected the manuscript as suggested.

22. The sentence in lines 218-221 could be clearer. (It is much clearer in the figure 4 legend.)

We rephrased this sentence as suggested

23. Figure 4:

a. In panel a, what I assume to be the GIS layer (grey leading edge) should be labelled. Consider also to adjust the lamella geometry to be more similar to Fig. 1g bottom (GIS layer and back trench not perpendicular to FIB).

We have updated the schematic in panel a as suggested

b. In panel c, why is the colour map for local resolution different from figure 2g?

Please see our reply to point 20a.

c. In panels d and e, how were the curves fitted?

We changed the curve fitting to be done with an exponential decay function, and reported the formula and the R2 values.

Discussion:

1. As mentioned above, consider speculating on the mechanism of the cleaning procedure and if it would work with other ion species.

We added the following sentence to the discussion: "We hypothesise that the mechanism for ice contamination removal could be due to changes in charging conditions of the ice due to exposure to ions, and can likely also be used with other ion species."

2. In line 276 and 330, a "~2 µm damaged area" is mentioned. In the results section it was referred to as "approximately 1 µm" in line 171 and "0.5 to 1.5 µm" in line 200. As mentioned above, I think a plot of the size distribution of these areas would be helpful.

We added a distribution of the length of the amorphous area between lamella in Supplementary Figure 7, and replaced the estimates with the exact values where appropriate, or removed the estimated.

3. As mentioned in the Introduction section, a small discussion of the advantages and disadvantages compared to "high-throughput" lift-out would help to put this advancement of the waffle method in context.

We added a short section in the discussion where we compare the relative advantages and disadvantages of serial liftout and waffle.

Methods

1. Please add the version number for Maps in line 376.

We added the version of Maps used to the manuscript.

2. Consider adding an image or the dimensions of the line pattern used for notch milling.

We added to panel 11 a schematic overview of the dimensions and angles used to prepare the notch.

3. I would like to encourage you to publish the autoTEM templates used for the shown lamellae with this manuscript.

We will upload this in the source data file + EMPIAR

4. In line 389, it is stated that the overviews were collected at 20 degrees tilt. Is this also the start tilt angle for the dose-symmetric tilt series acquisition?

Yes, we added this now to the methods section.

5. In line 399, a custom script for excluding tilts is mentioned. Does this script simply remove the corresponding blocks from the mdoc file, or does it have additional functionality?

This script removes the corresponding user-specified blocks from the mdoc file, renumbers the ZValue fields and updates the date/time stamp to be in the correct format for Warp.

6. The benchmarking of filters for particle picking is great and should be mentioned in the main text to make sure readers will come to the methods section to learn more.

We thank the reviewers for their kind comment and have included an explicit reference to this in the main text.

7. In line 423, does box size refer to a 64x64 image or a 64x64x64 volume?

Updated to 64³ voxels for clarity

8. In line 425, does "using the average" refer to the output of a 3D auto refine run or an average without alignment?

An average without alignment. We now clarify in the manuscript that it's a non-aligned average.

9. In lines 426-427, was the reference lowpass filtered or used at 16.2 Å?

The reference was lowpass filtered to 60Å, which we added to the manuscript.

10. In lines 432-435, do you have the data on how many tomograms were excluded due to each reason?

We did not record per-tomogram why they were excluded.

11. In line 437, do you have a guess as to what caused the software problems with these 13 tomograms?

This is the same problem we encountered during our work on argon (Berger, Dumoux and Glen *et al.* 2023). We still have not been able to find the root cause.

12. How was local resolution determined?

Local resolution for the consensus ribosome (Fig. 2g) was determined by M, local resolutions for surface damage particles (Fig. 4c, Supplementary Fig. 5) were determined by RELION local resolution jobs.

13. In lines 448-449, were the positions of tilt series found by eye or by cross-correlation/image registration?

The positions were found by eye, based on the first tilt-image and screenshots taken in tomo5 with the approximate position of each tilt-series. We clarified in the manuscript that the positions were manually found.

Supplementary Movies

1. Consider adding the total reconstructed z-height of the tomograms shown in Supplementary movie 2 to the legend.

We added the thickness of the four tomograms shown in movie 2. The tomograms shown in movie 2 have a Z-height of 171, but these have been rotated to correct for the 20 degrees milling angle and cropped. We added the reconstruction Z-height of the bin8 tomograms used for particle picking to the materials and methods section.

Data availability

1. I would like to encourage you to also deposit the raw data as well as the lamella overviews and grid map.

The EMPIAR accession code has now been added to the manuscript.

Co-reviewed by Fabian Eisenstein/Martin Pilhofer

Reviewer #2 (Remarks to the Author):

We thank the reviewer for thoroughly reviewing the manuscript and for all the helpful comments and suggestions.

Reviewer #3 (Remarks to the Author):

In this manuscript the authors describe application of Xenon plasma FIB milling in cryogenic conditions to generate lamellae from high-pressure frozen cells using an Arctis cryo-plasma FIB-SEM, of which they have an early commercial release. Generating thin lamellae is essential for in situ structural biology using cryo-EM and cryo-ET. Current approaches use a Gallium FIB which has the limitation that the relatively low current density makes it impractically slow when milling thick samples such as high-pressure frozen cells, limiting interrogation of tissue samples. In this paper the authors describe using a Xenon plasma source, which has the dual benefit of a higher current density, allowing higher currents to be used for milling, and a higher atomic mass, increasing the sputtering yield: effectively making milling faster.

To demonstrate this, the authors use high-pressure frozen E.coli samples. They describe some of the practical strategies that they needed to use such as removing ice contamination with a low magnification PFIB imaging and the angles and different currents used which will be of use to others wishing to try this approach. They show that they are able to generate a 4Å reconstruction of the ribosome using subtomogram averaging, and use subtomogram averaging of ribosomes at different depths from the lamella surface to measure damage from milling. The results from this study are aligned with expectations from simulations and from decades of work in the material sciences. This study will be useful to the field as a practical guide for the direct application to cryogenically frozen biological samples. The manuscript would benefit from more information about milling times.

We thank the reviewer for thoroughly reviewing the manuscript and for all the helpful comments and suggestions.

Comments below:

-In Figure 3 the authors note an amorphous area on the backside of the lamella that appears vitreous and without features. The authors propose that this is due to damage from the initial milling at 60nA. Since 1) there is a distinct barrier between this layer and the rest of the lamella, 2) it only appears on the backside and 3) there is an alternating pattern of high and low electron dense material can the authors rule out that this is redeposited material rather than damage to the extant lamella? The authors should consider alternate causes of this layer and not claim that it is the result of 60 nA milling unless they have a contrast with different currents.

To help determine whether 60 nA milling is indeed the cause of the amorphous and striated layers on the back of the lamellae, we performed additional experiments to compare perpendicular trench milling with 4 nA and 60 nA, and quantified the length of the amorphous areas (Supplementary Figure 7). Both conditions result in an amorphous area on the back of the lamellae, although with 4 nA it is significantly shorter in length. These results suggest that although the milling current appears to play a role, other factors also contribute to it.

We changed the manuscript accordingly in the light of these new data, and also suggest redeposition as a possible mechanism that may contribute to it.

-For Figure 4 I appreciate the use of matched controls and attempts to measure the damage curve rather than define a specific cut off for damaged/undamaged. This more rigorous step was not performed in their previous analysis of Argon FIB milling damage, so a direct comparison is difficult. The authors should be more cautious in their conclusion that Xenon and Argon have similar damaging propensities.

We thank the reviewer for their comment and appreciate the need for caution in comparison. In our previous study of the argon damage layer we indeed also used matched controls. We did not however plot a damage curve as is shown in Figure 4d. To reflect this, we have updated the figure to show a comparison (now Figure 4f) between argon and xenon. The curve shows that the fall off of the B-factor ratio to a value of 1 (i.e. no difference between the damage group and its corresponding matched control) occurs at a similar depth of ~ 40-45 nm from surfaces. We have reflected this in the text.

-Since one of the major arguments for using Xenon FIB milling is the potential time improvement, the authors should include a time estimate for each of the milling steps/ overall time to get that number of lamellae (and potentially contrast with Argon and Gallium). This could be included in Figure 2A. This result would be of great interest to the field.

Typically, it takes 1-1.5hrs to mill ~30 perpendicular trenches and 1-1.5hrs to clear material below these sites. The time range here is due to thickness variability between individual waffled grids. Subsequent thinning of each site takes on average 45 minutes, although this can be shortened with milling recipe optimisation. We have updated Figure 2A to include these overall time estimates for each major stage of the workflow, and note that Table 1 also contains estimates for each individual milling step. We have typically produced ~5000 µm² surface area of lamellae in a 24 hour session.

-What is the function of the curve fit in figure 4e and f? and what is the accuracy of this fit?

We changed the curve fitting to be done with an exponential decay function, and reported the formula and the R² values.

REVIEWER COMMENTS

Reviewer #1 (Remarks to the Author):

The authors added a considerable amount of data from an additional biological sample to the manuscript and demonstrated reproducibility of their presented workflow.

All our concerns were addressed and especially the comparison between the damage layer using 4 nA and 60 nA added an important piece to the data.

The presentation is more clear and contains additional statistics supporting the results. Therefore, we recommend the publication of this improved manuscripts pending some minor text revisions:

*Line 143: “lalla” typo

*Lines 302 and 314: duplicated paragraph

*Lines 417-418: disjointed sentence

We thank the reviewers for their comments and have made the relevant textual changes.

Reviewer #2 (Remarks to the Author):

Reviewer #3 (Remarks to the Author):

Overall I think that the paper is reporting a useful proof of principle: that lamella milled with Xenon can be used to generate subtomogram averages, and one suggested strategy to do so. In continuance of my earlier points, I think that any quantitative claims need to be backed up with statistics or de-emphasized.

We thank the reviewer for their critical analyses of the revised manuscript.

Thank you to the authors for now ambiguating about the cause of the amorphous layer on the backside of the lamella. I would encourage the authors to limit their emphasis on this layer. I would focus on the absence of evidence of devitrification, because this one of the concerns about increasing current, the ability to access higher currents is a key advantage of Xenon.

We changed the discussion (lines 286-309) on possible causes of the amorphous layer on the backside of the lamella to more strongly emphasise the absence of devitrification and place it in the context of concerns in the community that this may occur at higher currents. We think it is valid to place quite a substantial emphasis on the

amorphous layer, as the use of 60 nA for lamella fabrication or the presence of such a layer has not been reported before, and the risk for devitrification or other forms of damage at high currents are long-standing discussions in the cryo-EM community.

I maintain that the observed pattern is inconsistent with damage to the extant lamella and consistent with redeposition during milling. Whether there is damage closer to that edge of the lamella (i.e.: at <1 micron) was not assessed. The striations would be an expected result of redeposition when milling through the GIS layer and the cellular material resulting from the back milling strategy.

While we agree that redeposition is indeed a possible mechanism of formation of the amorphous layer, we maintain that we do not have sufficient evidence to exclude amorphisation as a possible mechanism. Our discussion reflects both of these options and highlights that they both depend on milling angle and current, which is a key difference with this milling protocol vs 'standard' lamellae preparation (60 nA perpendicular milling step). We have added the following sentence to explain how redeposition could explain the similarity in contrast between the amorphous layer and the biological sample:

'Since the back of the grid is not coated with a protective platinum layer, redeposited material from milling through the back would primarily be derived from the biological sample, which would explain the similar contrast observed in the amorphous layer.'

Our B-factor analysis did not reveal that proximity to the amorphous layer had detectable adverse effects on STA resolution, including in particles 0-1 μm from the amorphous layer. However, it was not possible to assess the effects of proximity with sampling of distances finer than 1 μm as the number of tomograms the particles per bin are derived from would become too low. Although we cannot exclude that within the 0-1 μm layer there is damage localised closer to the back edge, any possible damage is unlikely to have a significant effect on overall STA, which is why we recommend to still collect data in these areas. We rephrased the sentence in the discussion on this (line 292-294) to further clarify that no effects were detected for proximity on the micron scale.

How the authors defined the depth of damage remains unclear. I did not find a statistical or quantitative justification for making this call. In figure 4d and corresponding sup figs, the B-factor ratios are close to 1 at ~ 30 nm. However, there is some variation in these exact B-factor ratios (not unexpected) and, consequently, it is unclear what the built in error in their measurement is. (Perhaps this could be estimated from the variation in the "matched control" sets?) It is unclear how they conclude a depth of 45 nm from these data. Are the authors using the point at which the fitted curve crosses $y=1$? In the figure 4f visually the fitted Xenon curve does not cross 1 in the shown range and the Argon curve crosses at ~ 42 nm, if you solve the listed equations for $y=1$ you get different values: Figure 4f: Argon $y=1$ at 49nm, Xenon $y=1$ is undefined. Are the authors sure that the equations listed matches the curves shown? From the equation in 4d, Xenon $y=1$ at 50nm. The curve fits have a reported r^2 of 1, despite one of the values clearly not fitting the curve in figure 4d, and the fitted equation varies between experiments. This suggests to me that this study is underpowered to define the

depth of damage beyond a statement that there is appreciable damage within 30nm of the lamella surface.

As the magnitude of damage effect is expected to progressively fall off with increasing distance from the surface, we attempt to describe the damage curve rather than defining a cutoff of damage penetration, as recognised by the reviewer's comment in the previous revision.

The 30 to 45 nm depth of the damage is intended as a range in which the damage becomes negligibly small, and we have updated references to these values in the manuscript to emphasise this nuance. We have removed references to a 'damage layer' to avoid implication of a discrete surface depth from which there would be no more damaging effects of FIB milling. We have also added the following sentences in the discussion:

'Given the stochastic nature of surface damage propagation from ion-sample interactions, the effects of ion beam damage on structural information are expected to decay exponentially with increasing distance from lamella surfaces, thus caution is required in defining a limit to the depth of damage penetration. We therefore use B-factor ratios between each depth group and its corresponding matched control to describe the damage curve.'

We agree that there is inherent variation in the calculation of B-factors due to the random sampling of particle subsets, and so we have repeated all analyses in the manuscript such that there are now $n=3$ resolution estimates for any given number of particles used for the initial B-factor calculations. The gradient of $\ln(\# \text{ of particles})$ vs $1/\text{resolution}^2$ has been refitted for all particles with least squares regression and we report the standard errors. The updated B-factor ratios in Fig. 4d are now shown with the propagated standard error of the given ratio, and the 3-parameter exponential curve has been refitted so that it is weighted for these uncertainties in the B-factor ratios. With this fit there is no solution to $y=1$, however the values of the ratio are e.g. 30nm = 1.033, 35nm = 1.015, 40nm = 1.009, 45nm = 1.006. Due to the asymptotic nature of this depth/damage function, there is inherent uncertainty in precisely defining the value at which this asymptote is effectively reached. As there is no discrete cutoff for the depth of damage penetration, we prefer to describe the effects of damage becoming increasingly negligible/undetectable by up to approximately 45 nm.

Because of these additional steps we have taken to account for the variance in the calculation of the B-factors compared to our previous work using argon, we agree with your previous point to be more cautious in comparing xenon and argon directly, and therefore removed the panel added during the previous revisions where we try to make that comparison. We have also added the following text in the discussion:

'The way in which different studies define the depth to which damage penetrates also varies, which convolutes the ability to directly compare reported relative damage penetration between ion sources.'

The fitted curve equations have too many significant figures for the data shown and the text in the equations shown are too small to read all the terms.

We have updated the fitted curve equations to have 3 significant figures and larger text.

The figure in 4e allows them to say only that the group of particles <45nm from the surface and those >45nm from the surface appear to have different B-factors, not that the damage extends to 45nm. The data points do not appear to be in a linear range.

In the manuscript we only use this figure to demonstrate the overall impact of damage in the different regions of the volumes on STA resolution, not to claim that this demonstrates the limit of the damage penetration. In recognition of the fact that the most substantial damage is within 30 nm of the surface, we have updated Fig 4e to compare particles <30 nm, >30 nm, and all particles (compared to 45 nm previously). We thank the reviewer for their point about the linear range and have refitted the B-factor for particles ≥ 1600 to ensure a linear fit here.

Lines 302 to 325 in the discussion contain repeated content.

We have removed the duplicated paragraph.

It is not relevant to the conclusions in paper but may be for future biologists who may want to use the data after deposition in EMPIAR: the images of yeast look unusual, what strain was used?

The yeast is a commercial baking yeast optimised for French bread baking (exact strain unknown). We have updated the manuscript to refer to reflect this in the methods.